# Overview of Bacterial Protein Toxins from Pathogenic Bacteria: Mode of Action and Insights into Evolution

**DOI:** 10.3390/toxins16040182

**Published:** 2024-04-08

**Authors:** Michel R. Popoff

**Affiliations:** Unité des Toxines Bactériennes, Institut Pasteur, Université Paris Cité, CNRS UMR 2001 INSERM U1306, F-75015 Paris, France; popoff2m@gmail.com

**Keywords:** bacteria, toxins, mechanism of action, superantigen, enterotoxin, neurotoxin, pore-forming toxin

## Abstract

Bacterial protein toxins are secreted by certain bacteria and are responsible for mild to severe diseases in humans and animals. They are among the most potent molecules known, which are active at very low concentrations. Bacterial protein toxins exhibit a wide diversity based on size, structure, and mode of action. Upon recognition of a cell surface receptor (protein, glycoprotein, and glycolipid), they are active either at the cell surface (signal transduction, membrane damage by pore formation, or hydrolysis of membrane compound(s)) or intracellularly. Various bacterial protein toxins have the ability to enter cells, most often using an endocytosis mechanism, and to deliver the effector domain into the cytosol, where it interacts with an intracellular target(s). According to the nature of the intracellular target(s) and type of modification, various cellular effects are induced (cell death, homeostasis modification, cytoskeleton alteration, blockade of exocytosis, etc.). The various modes of action of bacterial protein toxins are illustrated with representative examples. Insights in toxin evolution are discussed.

## 1. Introduction

Among the huge number of bacterial species, only a few of them are pathogenic for humans and animals and are responsible for mild to severe diseases. Bacterial pathogenicity is mediated by various factors. Two main types of pathogenicity factors can be considered, those that promote bacterial colonization and invasion of the host leading to infection and disease such as virulence factors involved in adhesion to host cells, invasion of non-phagocytic cells, intracellular motility, resistance to host defenses, and those which cause host cell damages, notably toxins. Bacterial toxins include protein toxins (exotoxins), which are secreted mainly through the Sec pathway in the external medium and endotoxins, which are bacterial wall constituents, mainly lipopolysaccharides. Thus, toxigenic bacteria secrete specific poisons, called toxins, which are responsible for specific lesions and symptoms. Bacterial protein toxins diffuse locally at the bacterial colonization site and are transported to various sites or organs. In contrast to toxigenic bacteria, invasive bacteria first adhere to host target cells and then inject directly into cells virulence factors (also called effectors), which trigger an endocytosis mechanism and modify other cellular functions, thus facilitating the entry of the pathogen into the cell and subsequent persistence. Bacteria exploit various secretion pathways to export macromolecules. The type III secretion system, also called injectisome, is the most common secretion system used by Gram-negative invasive pathogens to deliver virulence factors directly into host cells [1,2,3]. Although bacterial protein toxins and virulence factors of invasive bacteria might share similar enzymatic activities, bacterial protein toxins retain the unique properties to diffuse in the extracellular space and to recognize specific cell surface receptors. Bacterial protein toxins are either active at the cell surface or enter cells and interact with the intracellular target(s) (Figure 1). This review is focused on the various action mechanism types of bacterial protein toxins through representative examples.

## 2. Interaction of Toxigenic Bacteria with the Host

The habitat of most toxigenic bacteria is the environment. The first line of host defense is the integrity of tegument and mucosa which line the compartments exposed to the environment (digestive, respiratory, and uro-genital compartments). The oral-gastro-intestinal tract is the most exposed compartment to environmental bacteria and pathogenic bacteria have developed multiple modes of interaction with the host barriers. Interactions of bacterial pathogen–host with the intestinal mucosa and gas gangrene pathogenesis are schematically presented (Figure 2).

### 2.1. Cross Talk between Bacterial Toxins/Toxigenic Bacteria and the Intestinal Mucosa

Bacteria from the environment, notably those ingested with food, usually do not colonize the digestive tract of a healthy host since the resident microflora (gut microbiota) prevents the growth of exogenous bacteria. The intestinal tract is a complex ecosystem that contains up to 10^11^ bacteria per gram in the colon of humans and several hundreds of species. A crucial function of the gut microbiota is the barrier effect against non-established bacteria [4,5,6,7]. Pathogens use various strategies to overcome the host defenses.

Certain toxigenic bacteria can grow and produce toxins in food. The ingestion of preformed toxins in food, referred to as intoxination, induces specific diseases such as food-borne botulism, staphylococcal food poisoning, and *Bacillus cereus* emetic syndrome (Figure 2A).

Alteration of the gut microbiota (dysbiosis) is an appropriate condition for growth and toxin production in the intestinal content by some enteric pathogens. A well-documented example is antibiotic-associated diarrhea due to *Clostridioides* (previously *Clostridium*) *difficile*. Antibiotics perturb the gut microbiota composition and function inhibiting the barrier effect against *C. difficile*. Thus, this pathogen can grow and synthesize potent toxins (toxin A (TcdA) and toxin B (TcdB)) and the additional binary toxin (*C. difficile* transferase) according to the strains), leading to intestinal lesions and inflammation [8,9,10,11]. In *Clostridium perfringens* food poisoning, massive ingestion of bacteria allows the colonization resistance of the gut microbiota to be overcome and facilitates the multiplication and sporulation of *C. perfringens* in the content of the small intestine. Indeed, foods at risk of food poisoning contain at least 10^5^ enterotoxigenic *C. perfringens* per gram but rarely preformed *C. perfringens* enterotoxin (CPE). CPE is synthesized during the sporulation phase, which occurs in the intestinal tract and not in food and it is released subsequently to bacterial wall lysis in the intestinal content. Thus, the symptoms of *C. perfringens* food poisoning (diarrhea and abdominal pain) occur 8–24 h after the ingestion of contaminated food by *C. perfringens*, later than in the intoxination process of staphylococcal food poisoning (2–6 h), which is due to ingestion of preformed staphylococcal enterotoxins in food [12,13,14]. Another example of toxigenic bacteria growing in the intestinal content is provided by infant botulism. This disease occurs in newborns and infants up to one year old. A not fully developed or non-yet fully functional gut microbiota is a prerequisite condition for the growth of *Clostrium botulinum* and toxin production in the intestine, the contamination of which is mainly from the environment (Figure 2A) [15,16,17].

Colonizing bacteria express specific attachment factors to enterocytes and colonize the intestinal mucosa surface, which is one of the escape mechanisms to the colonization resistance mediated by the microbiota mainly localized in the gut lumen [6,18]. Some of them (*Vibrio cholerae*, enterotoxic *Escherichia coli*, …) produce potent enterotoxins responsible for intestinal disorders. Enteroinvasive bacteria use VFs injected into cells by type 3 secretion system (*Shigella*, enteroinvasive *E. coli*, *Salmonella*, …) or associated with the bacterial wall (*Listeria*) to promote their uptake into intestinal cells [19,20]. *Salmonella typhi* and *Salmonella typhimurium* invade the intestinal mucosa via M cells, epithelial cells, and macrophages and then can disseminate in lymph nodes and subsequently to various organs such as the liver and spleen (Figure 2A) [21,22]. The intracellular lifestyle of these pathogens facilitates their resistance to the host immune defenses.

### 2.2. Bacterial Toxins/Toxigenic Bacteria and Gas Gangrene

A characteristic model of bacterial gangrene pathogenesis is provided by clostridia such as *C. perfringens*. Clostridia are not invasive bacteria for healthy cells, they can enter a host through a wound. Local tissue destruction and hypoxia consecutive to the wound facilitates clostridium growth and subsequent toxin production such as *C. perfringens* α-toxin, perfringolysin (PFO), and hydrolytic enzymes (proteases, collagenases, hyaluronidases, etc.) (see below). Extension of tissue degradation by toxins allows the progression of bacterial growth and additional production of toxins. PFO and α-toxin induce a leucostasis and impair macrophage phagocytosis, resulting in the absence of tissue inflammatory response. Moreover, α-toxin causes platelet aggregation, obstruction of blood vessels, and subsequent tissue hypoxia. When higher amounts of toxins are produced, they are absorbed in the blood circulation and cause toxic shock and multi-organ failure [23,24,25] (Figure 2B).

## 3. Diversity in Bacterial Protein Toxins

Bacterial protein toxins exhibit a very wide diversity in molecules regarding their size, amino acid sequence, structure, and mode of action. They are single-chain proteins or multiproteins resulting from the non-covalent assembly of several proteins (binary, ternary, or more complex structures) with sizes ranging from small peptides (18–19 amino acids, *Escherichia coli* heat-stable enterotoxins (STs)) to large proteins (2710 amino acids for *C. difficile* toxin A (TcdA), 3500–5300 amino acids for multifunctional-autoprocessing repeats-in toxins (MARTX)) [26,27,28] (Figure 3). Single-chain protein and binary toxins are mostly produced by Gram-positive bacteria, while the other more complex multiprotein toxins are largely synthesized by Gram-negative bacteria.

Bacterial protein toxins contain distinct functional domains that drive their successive steps of activity in target cells. The first step of all bacterial protein toxins is the recognition of a cell surface receptor, which is mediated by a receptor binding domain, termed B domain or B subunit or component. Diverse cell membrane molecules are used as toxin receptors, proteins, glycoproteins, and glycolipids, which determine the toxin tropism such as neurotoxins for neuronal cells, enterotoxins for intestinal epithelial cells, leucotoxins for lymphocytes, hemolysins for red blood cells, and cytotoxins for a wide range of target cells. Many bacterial protein toxins are active through enzymatic activity and contain an enzymatic (A) domain or subunit. These toxins are referred to as AB toxins [29]. Intracellularly active toxins have additional domains, namely the translocation (T) domain, which facilitates the passage of the A domain into the cytosol, and the autoprocessing domain involved in the release of the A domain from the B domain. Most of the multiprotein AB toxins retain a complex structure, for example, one A subunit and five B subunits assembled in a pentamer (AB_5_: cholera toxin (CT), shiga toxin (ST)), two distinct A subunits and five B subunits (A_2_B_5_, typhoid toxin (TyT)), one A, and two B subunits (AB_2_, cytolethal distending toxin (CDT)). Pertussis toxin (PTX) has an AB_5_ structure with four distinct proteins in the B pentamer [29] (Figure 3).

**Figure 3 toxins-16-00182-f003:**
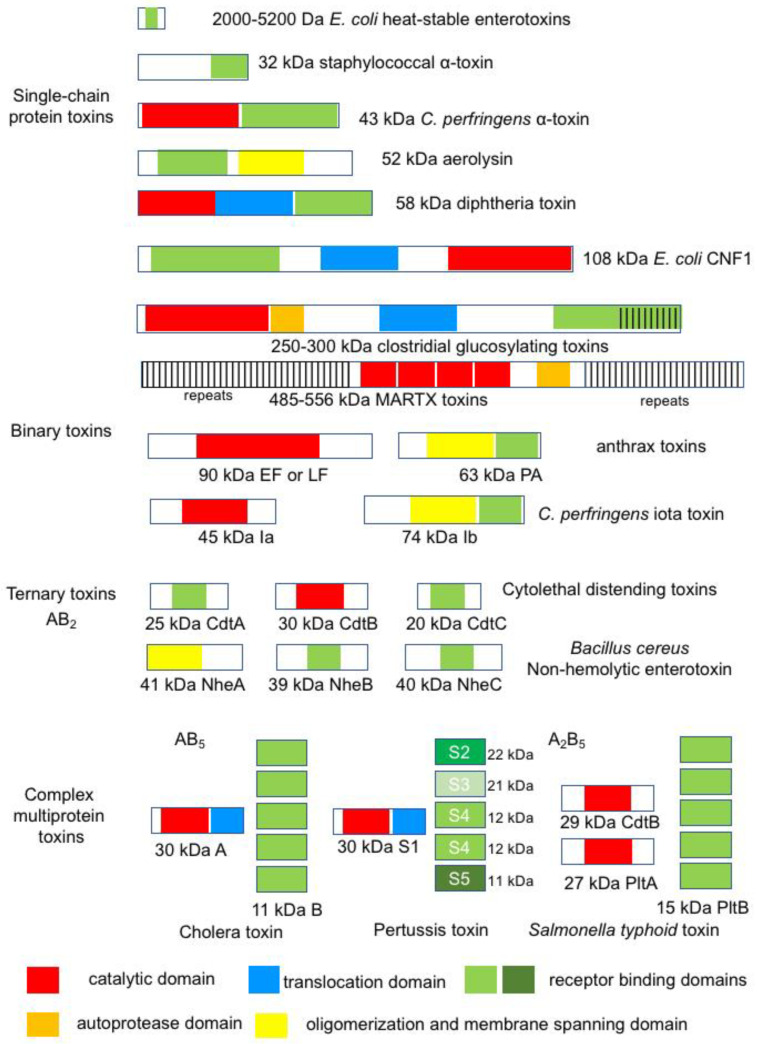
Diversity in size and structure of bacterial protein toxins through representative examples. All bacterial protein toxins contain a receptor-binding domain (green). Intracellularly active toxins have a translocation (blue) and catalytic (red) domain and some of them have an autoprocessing domain (orange). Pore-forming toxins share oligomerization and membrane-spanning domains (yellow).

Once bound to cell surface receptor(s), two main classes of bacterial protein toxins can be distinguished: those active on the cell membrane (signal transduction, membrane damage by pore formation, or hydrolysis of membrane compound(s)) and those active intracellularly. According to the intracellular target recognized by the A subunit of intracellularly active toxins and the type of modification, various cellular effects are observed (apoptosis, homeostasis, cytoskeleton alteration, blockade of exocytosis, etc.) (Figure 4).

Bacterial toxins exploit cellular mechanisms to enter the target cells. Two main endocytic pathways are used: a short pathway through early endosomes (EEs) and upon endosomal acidification translocation of the A subunit into the cytosol and a long pathway including EE/late endosomes (LEs), Golgi, endoplasmic reticulum (ER), and ER secretion system such as Sec61/63 (Figure 4) [29,30]. Most single-chain protein toxins and binary toxins deliver their enzymatic domain or enzymatic component through the endosomal membrane of acidified EEs, whereas AB_5_ toxins, the B pentamer of which recognizes gangliosides or other glycoconjugates as receptors, use a long endocytic pathway through EE, LE, Golgi, and ER [29].

Interaction between bacterial toxin and cell receptor not only drives the cell specificity of toxins but also promotes membrane curvature that is required for membrane invagination and formation of endocytic vesicles. Most toxins that enter cells via a short pathway recognize cell membrane proteins as a receptor and use clathrin-dependent endocytosis. Polymerization of clathrin associated with the activity of accessory proteins triggers membrane invagination and the formation of clathrin-coated vesicles. Bacterial toxins, which use a long pathway such as cholera toxin (CT) and Shiga toxin (ST), interact with glycosphingolipids as receptors, namely GM1 and globotriaosylceramide, respectively (see below). Their pentameric structure of B components binds to multiple receptor molecules leading to membrane lipid reorganization and clustering of toxin–receptor complexes, which promote curvature and tubular membrane invagination [31,32,33].

**Figure 4 toxins-16-00182-f004:**
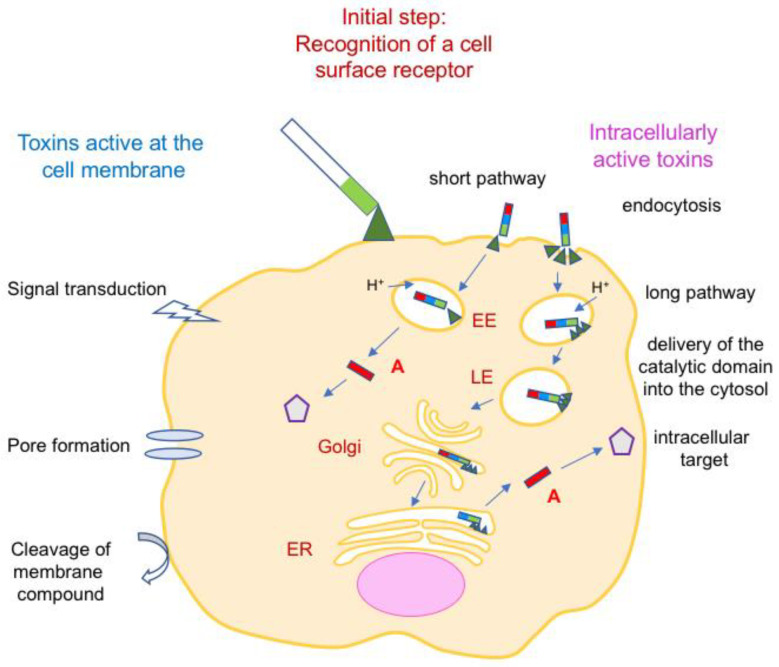
Main types of activity of bacterial protein toxins. First, toxins recognize cell surface receptor(s), which determine their cell specificity (neurotoxins, enterotoxins, leucotoxins, and cytotoxins). Toxins active at the cell membrane use either a transduction, pore formation, or enzymatic hydrolysis mechanism. Intracellularly active toxins exploit short or long endocytic pathways to internalize the catalytic (A) subunit, which interacts with an intracellular target and mediates the cellular effects. Interaction of intracellularly active toxins with cell surface receptors not only drives the cell specificity but also initiates membrane curvature and endocytic vesicle formation. EE, early endosome; LE, late endosome; ER, endoplasmic reticulum.

## 4. Bacterial Protein Toxins Active at the Cell Membrane

### 4.1. Toxins Transducing a Transmembrane Signal and Hormone-Like Toxins

A typical example of bacterial toxin acting as a hormone-like is the thermostable enterotoxin type A (STa) of enterotoxigenic *E. coli*. STa is responsible for diarrhea in children and adults, such as travelers’ diarrhea and endemic diarrhea in developing countries, as well as in animals. Mature STa from human *E. coli* strains comprises 19 amino acids including 6 cysteines forming 3 intramolecular disulfide bonds. This compact structure confers heat resistance. STa shares a similar structure and 50% amino acid sequence homology with the hormone guanylin, which is produced in the intestine and regulates water and electrolyte fluxes in the gut. STa recognizes the intestinal guanylyl cyclase C (GC-C), a membrane enzyme of epithelial cells of small intestine and colon, as a receptor. Binding of STa to the extracellular domain of GC-C leads to GC-C dimerization and subsequent activation of the intracellular catalytic domain. This results in hydrolysis of guanosine 5′-triphosphate (GTP) and excessive formation of cyclic GMP (cGMP), which stimulates the protein kinase II and subsequently the phosphorylation of the Cl^−^ channel called cystic fibrosis transmembrane conductance regulator (CFTR). In addition, cGMP inhibits the phosphodiesterase 3 (PDE3) leading to the accumulation of cAMP, activation of protein kinase A (PKA), additional activation of CFTR, and inhibition of the Na^+^ channel (Na^+^/H^+^-exchanger 3, NHE3), impairing the reabsorption of Na^+^ (Figure 5). Thus, STa mimics the role of guanylin by activating Cl^−^ HCO3^−^ and fluid secretion in the intestinal lumen but in contrast to guanylin, the effects of STa are irreversible and more potent. The studies of STa allowed the development of novel drugs modulating Cl^−^ channels and novel strategies for the treatment of diarrheal diseases and colorectal cancer [28,34,35].

### 4.2. Toxins Superantigens

Superantigens (SAgs) are a family of potent mitogenic toxins produced mainly by *Streptococcus* spp. (notably *S. pyogenes*) and *Staphylococcus aureus*. They are 20–30 kDa single-chain proteins. Staphylococcal SAgs encompass staphylococcal enterotoxins (SEs, at least 25 types SEA to SElZ and multiple variants), which are responsible for food-borne poisoning and the toxic shock syndrome toxin (TSST). Eleven SAgs have been identified in *S. pyogenes*, which is involved in skin and soft tissue infections as well as in toxic shock syndrome. Additional SAgs have been reported in *Yersinia pseudotuberculosis* and *Mycoplasma arthritidis* [36].

SAgs bind simultaneously to two cell surface receptors on immune cells, major histocompatibility complex (MHC) type-II molecules on antigen-presenting cells and the T cell receptor (TCR) on lymphocytes T. SAgs retain a conserved structure, which consists of two domains containing binding sites separated by a central α-helix [36]. Thus, SAgs bypass the conventional antigen presentation, which consists of antigen phagocytosis, processing and peptide expression with MHC class II molecules on the antigen-presenting cell surface. Since SAgs recognize variable regions of TCR, up to 25% of T cells are activated compared to 0.001–0.0001% T cell activation in the conventional peptide presentation (Figure 6). The intense proliferation of both CD4 and CD8 T cells lead to massive release of pro-inflammatory cytokines such as interleukin 1-β, interleukin-2, and tumor necrosis factor α, which are responsible for fever and shock [36,37,38,39].

Staphylococcal food poisoning is characterized by nausea, vomiting, abdominal cramping, and diarrhea occurring within 3–9 h after ingestion of food containing preformed SEs. The potent emetic activity of SEs is due to their ability to pass across the intestinal epithelium and to interact with mast cells in the lamina propria or in submucosa through unknown receptors and unknown mechanisms, leading to the release of 5-hydroxytryptamine (serotonin), which activates the vagus nerve and subsequently the emetic center of the central nervous system [40,41,42].

### 4.3. Membrane-Damaging Toxins

Bacterial membrane-damaging toxins encompass pore-forming toxins (PFTs) and enzymatically active toxins against membrane compounds. They are the main toxins of bacteria involved in gangrene/myonecrosis and soft tissue infections. Certain PFTs interact specifically with the intestinal or lung epithelial barrier leading to specific diseases.

#### 4.3.1. Bacterial Pore-Forming Toxins

Pore formation through the cell membrane is the most common mechanism of bacterial toxins. About 30% of bacterial toxins are PFTs [43]. PFTs can be divided into two main classes, α-PFTs and β-PFTs, according to their structure-based interaction with the lipid bilayer (Table 1). PFTs are secreted as soluble monomers, which bind to specific receptors on cell membranes, oligomerize, and form transmembrane pores. α-PFTs are rich in α-helices and use a bundle of amphipathic α-helices to build the transmembrane pore (Figure 7A and Figure 8). Cytolysin A (or hemolysin E) from *E. coli* is the prototype of the Cytolysin A family, which encompasses related hemolysins from *Salmonella enterica* and *Shigella flexneri*. Cytolysin A monomers show two domains, a head domain containing a small β-hairpin (β-tongue) and a tail domain consisting of five α-helices. Cytolysin A spontaneously oligomerizes into pore complexes in solution in the presence of detergent or in membrane. Assembly of the monomers in the dodecamer pore induces a conformational change. Notably, the β-tongue adopts an α-helical structure and the N-terminal helix swings upward. The bundle of the dodecameric N-terminal helices inserts into the membrane forming a mainly hydrophilic pore [44,45].

Repeat-in-ToXin (RTX) toxins constitute a vast toxin family that is produced by numerous Gram-negative bacteria. Representative RTX toxins are *E. coli* α-hemolysin (HlyA), *Actinobacillus* leucotoxins, *Pasteurella* leucotoxins, and *Bordetella pertussis* adenylate cyclase (CyaA) (Table 1). RTX toxins are single-chain proteins of 100–200 kDa, which contain glycine- and aspartate-rich nonapeptide repeated sequences in the C-terminal part. Calcium binding to the nonapeptide repeats is required for the acquisition of a functional conformation of RTX toxins. In addition, RTX toxins are activated through acylation at one or two lysines in the central region of the toxins. After binding to cell surface receptor(s), RTX toxins oligomerize and insert amphipathic N-terminal α-helices in the cell membrane that form a pore [46,47,48,49]. CyaA is distinct from the other RTX toxins by containing an N-terminal catalytic domain, which is an adenylate cyclase. CyaA is a major toxin of *B. pertussis* that contributes to its colonization by impairing the phagocytosis function in neutrophils and macrophages. CyaA uses the pore formed by the RTX domain to internalize the catalytic domain. Thus, CyaA promotes massive levels of cAMP and pores through the plasma membrane leading to inhibition of cellular functions and apoptosis [47,49,50,51]. A subfamily of the RTX toxins encompasses the MARTX toxins. MARTXs have been first identified in *Vibrio* sp. (*V. cholerae* and *V. vulnificus*) and then in numerous bacterial genera of Gram-negative bacteria such as *Aeromonas*, *Chromobacterium*, *Photorhabdus*, *Proteus*, *Xenorhabdus*, and *Yersinia*. They are very large single-chain proteins of 3500 to 5300 amino acids, which contain four regions: N-terminal repeats, central effector domains, cysteine protease domain, and C-terminal repeats. The repeat regions are involved in pore formation, which can lead to membrane disorganization and cell lysis. Pores also mediate the translocation into the cytosol of the central effector domains, which are processed by the cysteine protease domain. MARTX contains up to five cytopathic effectors such as alpha-beta hydrolase, actin crosslinking domain, adenylate cyclase, Rho-inactivating domain, Ras/Rap1 specific peptidase, and VIP2 (vegetative insecticidal protein2)-like protein, which impair cellular key functions leading to actin cytoskeleton alteration and cell rounding, inhibition of phagocytosis, inhibition of cell proliferation, and apoptosis [26,52,53,54].

*Bacillus thuringiensis* produces multiple insecticidal toxins including the three-domain parasporal crystal (3d-Cry) toxins, which are α-PFTs. Cry toxins are synthesized as 130 or 70 kDa proteins, which are solubilized and activated in the insect gut by proteolytic cleavage of N- and C-terminal parts. The Cry active form (65 kDa) is composed of three domains: domain II and III are involved in binding to receptors on insect intestinal cells and domain I, which contains seven a-helices, oligomerizes, and forms a channel through the cell membrane [55,56,57,58] (Figure 7A).

β-PFTs are divided into several families based on their structural conformation but they share a general mechanism of pore formation (Figure 7B and Figure 8). They contain at least three structural domains, namely receptor-binding, oligomerization, and pore-forming domains such as rim, cap, and stem domains of *S. aureus* α-toxin and related toxins. Most aerolysin family PFTs such as *C. perfringens* ε-toxin (ETX) retain a similar structure, except aerolysin, which contains an additional N-terminal domain (D1) involved in receptor binding with residues from domains 2 and 4 (Figure 7B). Cholesterol-dependent cytolysins (CDCs) and binding components of binary toxins show also four domains.

Secreted β-PFTs soluble monomers bind to cell membrane receptor(s) through the C-terminal domain in CDCs, hemolysin, or *S. aureus* α-toxin family, binding components of binary toxins and N- or C-terminal domains in the PFT aerolysin family. Mobility of monomers bound to receptors on the membrane facilitates the interaction between monomers and subsequent oligomerization. Indeed, the main receptors of β-PFTs (cholesterol, sphingomyelin, and glycophosphatidylinositol (GPI)-anchored proteins) are localized in membrane microdomains such as lipid rafts capable of lateral mobility [59]. Oligomerization through interactions between the other domains leads to the formation of prepore, which extends on the membrane surface. Binding to receptor and oligomerization trigger a conformational change in the central domain leading to the unfolding of a short α-helix into an amphipathic β-strand. The amphipathic β-strands from each monomer assemble into an amphipathic β-barrel. The prepore collapses and the amphipathic β-barrel inserts into the membrane forming a transmembrane pore (Figure 8) [60,61,62,63,64].

CDCs recognize cholesterol as a receptor, assemble in large oligomers (30–50 monomers), and use two transmembrane hairpins from each monomer to form large pores (Table 1) (Figure 7) [60,65,66]. CDCs are essentially produced by Gram-positive bacteria and PFO is the prototype of the CDC family [60,66,67,68]. PFO causes cell lysis and is a major toxin involved in *C. perfringens* gangrene and myonecrosis [25]. In contrast to most CDCs, which are active at neutral and acidic pH and act extracellularly, Listeriolysin O (LLO) has an optimal pH of activity at 5.5 and thus LLO preferentially lyses the membrane of host acidified phagosome, leading to the escape of *Listeria* into the cytosol. Moreover, LLO-specific N-terminal sequences (proline, glutamic acid, serine, and threonine (PEST)-like sequence) contribute to the toxin compartmentalization in phagosomes [69,70]. In addition to their direct cytotoxic effects by creating large pores and membrane alteration, CDCs can translocate specific effectors into cells through permeabilized membranes. Thereby, streptolysin O (SLO) mediates the internalization of *S. pyogenes* NAD-glycohydrolase, which contributes to the cytotoxicity and to the induction of inflammatory response [71].

β-PFTs from the aerolysin and hemolysin or *S. aureus* α-toxin families generate small pores by forming small oligomers (hepta- or octamers) in which each monomer contributes by only one amphipathic β-hairpin in the structure of the β-barrel (Figure 7B). Staphylococcal leucocidins are bicomponent toxins that use two distinct classes of proteins, S and F proteins, to form hetero-octameric pores in a 1:1 stoichiometry [72]. Recognition of receptors via the receptor binding domain drives the cell specificity of β-PFTs (Table 1). For example, *S. aureus* α-toxin, which interacts with various cell types including epithelial cells, as well as leucocidins, which target leukocytes, play a major role in *S. aureus* infections [72,73]. *C. perfringens* enterotoxin specifically recognizes the enterocytes and is responsible for *C. perfringens* foodborne poisoning [74,75], whereas *C. perfringens* ε-toxin, which is produced in the intestine, passes through the intestinal barrier without significant alteration of intestinal cells, transits in the blood circulation, targets kidney cells, crosses the blood–brain barrier, and interacts with neurons and oligodendrocytes leading to glutamate release and subsequent neurological symptoms of excitation [76].

β-PFTs induce membrane permeabilization leading to multiple effects in target cells. Early events consist of massive loss of intracellular K^+^ and ATP and entry of Na^+^ and Ca^++^. Changes in ion composition trigger multiple cellular responses such as activation of certain proteases and phosphatases, activation of signaling cascades such as mitogen-activated protein kinases (MAPKs) and the inflammasome, changes in expression of certain genes, and cell death via necrosis or apoptosis (reviewed in [77,78,79]). For example, efflux of K^+^ by aerolysin triggers the formation of the NLRP3 inflammasome and activation of caspase-1 [80]. *C. perfringens* ε-toxin induces rapid depletion in K^+^ and ATP, leading to necrosis of renal cells [81,82]. High concentrations of *C. perfringens* enterotoxin promote Ca^++^ influx in intestinal cells and death by necrosis via activation of calpain and receptor-interacting serine/threonine-protein kinase-1 and -3 (RIP1 and RIP3) and assembly of mixed-lineage kinase domain-like pseudokinase (MLKL) [83].

Binding components of binary toxin such as the protective antigen (PA) of *B. anthracis* toxins share a similar four-domain structure with those of CDCs. However, they form small heptameric or octameric pores with only one amphipathic β-hairpin from each monomer in the β-barrel, similarly to aerolysin and *S. aureus* α-toxin (Figure 7B). Binding components play a major role in the internalization of the corresponding enzymatic components. Thereby, PA binds to cell surface receptors (capillary morphogenesis 2 (CMG2) and tumor endothelial marker 8 (TEM8)), which are localized in lipid rafts, oligomerizes, and traps either the edema factor (EF) or the lethal factor (LF), with one to five EF/LF molecules per heptamer or octamer. The whole assembled toxin is endocytosed in clathrin-coated vesicles. Upon acidification of EEs, the PA prepore inserts into the endosomal membrane, leading to a functional pore that allows the translocation of partially unfolded LF or EF into the cytosol [84,85,86,87,88,89,90]. The binding components, Ib and C2-II, of *C. perfringens* iota toxin and *C. botulinum* C2 toxin, respectively, recognize distinct cell surface receptors (Table 1) and use a similar mechanism as PA to translocate their enzymatic components, Ia and C2-I, respectively, into the cytosol from acidified endocytic vesicles [91,92].

**Table 1 toxins-16-00182-t001:** Bacterial pore-forming toxin families and representative pore-forming toxins.

Toxin Family	Toxins	Toxin Producing Organism	Receptor	Oligomers (Number of Monomers)	Pore Size	Disease/Activity	References
α-Pore-Forming Toxins							
Cytolysin A	Hemolysin E (or ClyA)	*Escherichia coli*	CD11/CD18 integrin	12	3.5 nm		[44]
	Non-hemolytic enterotoxin (NHeA) (tripartite toxin: NheA, NheB, NheC)	*Bacillus cereus*	cholesterol	?	~2 nm	food poisoning	[93,94,95]
	hemolysin BL (Hbl-B) (tripartite toxin: Hbl-B, Hbl-L1, Hbl-L2)	*Bacillus cereus*	LITAF, CDIP1	7–8	1–2 nm	hemolysis, enterotoxicity	[96,97,98]
RTX	α-hemolysin (HlyA)	*Escherichia coli*	CD11/CD18 integrin, glycophorin	?	1.1–2 nm	virulence factor associated with CNF in uropathogenic *E. coli*	[46,99,100]
	Adenylate cyclase (CyaA)	*Bordetella pertussis*	CD11b/CD18 integrin	variable 12?	0.6–0.8 nm	Translocation of the catalytic domain, whooping cough	[50,101,102,103]
	MARTX	*Vibrio vulnificus* *Aeromonas hydrophila*	?		1.8 nm	delivery of effector domains, intestinal tissue destruction	[26,53,104]
3d-Cry Toxins	Three domain Crystal (Cry) Toxins	*Bacillus thuringiensis*	cadherin, ABC transporter subfamily C2, aminopeptidase, alkaline phosphatase	4	4–5 nm	insecticidal activity	[55,56,57,58]
β-Pore-Forming Toxins							
Cholesterol-dependent cytolysins (CDCs)	Perfringolysin O (PFO)	*Clostridium perfringens*	cholesterol, glycans	40–50	25–45 nm	myonecrosis, gangrene	[66,105,106]
	Botunolysin	*Clostridium botulinum*	cholesterol	30–50		hemolysis	[65]
	Tetanolysin	*Clostridum tetani*	cholesterol	30–50		hemolysis	[65]
	Streptolysin O (SLO)	*Streptococcus pyogenes*	cholesterol, glycans	50–80	30 nm	hemolysis	[106,107]
	Listeriolysin O (LLO)	*Listeria monocytogenes*	cholesterol, glycans	30–50	30 nm	*Listeria* vacuolar escape	[106,108]
	Intermedilysin (ILY)	*Streptococcus intermedius*	cholesterol, CD59, N-linked glycan	30–50	25–30 nm	tissue destruction	[106,109,110]
	Pneumolysin (PLY)	*Streptococcus pneumoniae*	cholesterol, glycans	44	26 nm	pneumonia, meningitis, otitis	[106,111]
	Anthrolysin (ALO)	*Bacillus anthracis*	cholesterol, glycans	30–50			[65,106]
Aerolysin	Aerolysin	*Aeromonas* sp.	GPI-anchored proteins	7	0.7–1.7 nm		[112,113,114,115]
	ε-toxin (ETX)	*Clostridium perfringens*	HAVCR1, MAL	7	1–2.4 nm	animal enterotoxemia	[116,117,118]
	Enterotoxin (CPE)	*Clostridium perfringens*	Claudins	7	1.4 nm	food poisoning	[119,120,121,122]
	α-toxin	*Clostridium septicum*	GPI-anchored proteins	6, 7	1.2–1.6 nm	myonecrosis, gangrene	[123,124,125]
	Cry toxins of ETX-MTX subfamily, Mosquitocidal toxin (MTX)	*Bacillus thuringiensis* *Bacillus sphaericus*	insect gut receptor			insecticidal activity	[126]
Hemolysin or *S. aureus* α-toxin	α-toxin	*Staphylococcus aureus*	Phosphatidylcholine, sphingomyelin, ADAM10 disintegrin	6, 7	1.4–3 nm	skin necrosis, soft tissue infections	[127]
	Panton-Valentine leucocidin LukS-LukF	*Staphylococcus aureus*	C5aR	8	1.9–2.1 nm	necrotizing pneumonia	[72,128,129,130]
	γ-hemolysin (HlgA/HlgB-HlgC)	*Staphylococcus aureus*	CXCR1, CXCR2, CCR2, C5aR	8	2.5–3 nm	skin, soft tissue infections	[72,131,132,133]
	Leucocidin LukA-LukB (LukGH)	*Staphylococcus aureus*	CD11b, HVCN1	8	3 nm	soft tissue infections	[134,135,136]
	Leucocidin LukE-LukD	*Staphylococcus aureus*	CCR2, CCR5, CXCR1, CXCR2, DARC	8	1.9–2.1 nm	soft tissue infections	[137,138,139,140]
	Beta-toxin	*Clostridium perfringens*	Platelet endotheial cell adhesion molecule-1 (CD31)	8	1.2–2.0 nm	necrotic enteritis	[141,142]
	Net-B	*Clostridium perfringens*	cholesterol	7	1.6 nm	necrotic enteritis	[143]
	Delta-toxin	*Clostridium perfringens*	monosialo-ganglioside (GM2)	7	4 nm		[144,145]
	CctA	*Clostridium chauvoei*				myonecrosis, blackleg	[146]
	*Vibrio cholerae* cytolysin (VCC)	*Vibrio cholerae*	Glycoconjugates	7	2.5 nm	hemolysis enterotoxicity	[147,148,149]
	*Vibrio vulnificus* hemolysin	*Vibrio vulnificus*	gangliosides, N-acetyl-D-galactosamine, N-acetyl-D-lactosamine	7		apoptosis	[150,151]
Binding components of binary toxins	Protective Antigen (PA)	*Bacillus anthracis*	capillary morphogenesis protein 2 (CMG2), tumor endothelial marker 8 (TEM8)	7	1.2 nm	Translocation of the enzymatic components EF, LF	[89,90,152,153,154]
	Iota toxin B component (Ib)	*Clostridium perfringens*	lipolysis-stimulated lipoprotein receptor	7	1 nm	Translocation of Ia	[92,155,156]
	C2 toxin B component (C2-II)	*Clostridium botulinum* C and D		7	1–2 nm	Translocation of C2-I necrotic enteritis	[116]
	*Clostridium difficile* transferase (CDTb)	*Clostridioides difficile*	lipolysis-stimulated lipoprotein receptor	7		Translocation of CDTa pseudomembranous colitis	[155,157]
	Vegetative insecticidal protein B component (VIP1)	*Bacillus thuringiensis*	insect midgut membrane receptor	7?		Translocation of enzymatic component	[158,159]

MARTX, multifunctional autoprocessing repeats-in-toxin 10; CctA, *Clostridium chauvoei* toxin A; DARC, Duffy Antigen Receptor for Chemokines; LITAF, LPS-induced TNF-α factor; CDIP1, cell death involved p53 target 1; HVCN1, human voltage-gated channel 1.

**Figure 7 toxins-16-00182-f007:**
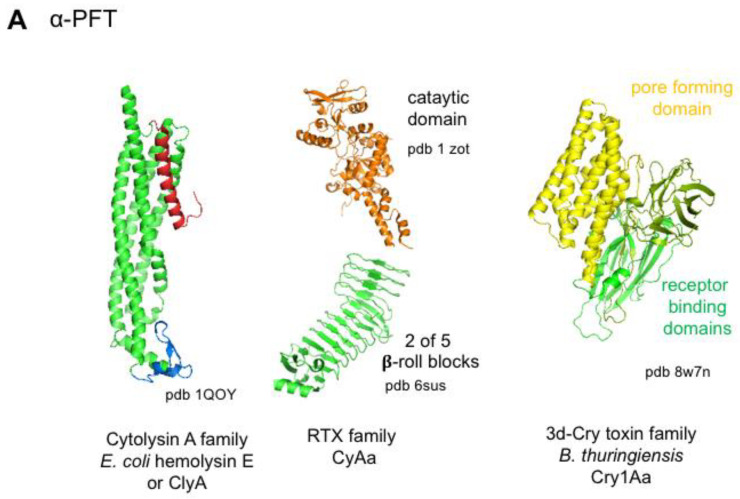
Structures of representative α- and β-pore-forming toxins (PFTs) from the main PFT families. (**A**) Structures of soluble monomers of representative α-PFTs: *E. coli* hemolysin E or cytolysin A (ClyA), *Bordetella pertussis* adenylate cyclase (CyAa) (partial structure) from the RTX family, and *B. thuringiensis* Cry1Aa from the 3d-Cry toxins. (**B**) Structures of representative toxins from each β-PFT family showing soluble monomers, protomers from complex pores, and pores. *Streptococcus peumoniae* pneumolysin from the cholesterol-dependent cytolysins (CDCs). Each CDC monomer contains two α-helices, which change conformation into trans-membrane β-hairpin (TMH) upon oligomerization to form the β-barrel. Aerolysin contains an additional N-terminal domain (D1), which is missing in the other aerolysin family PFTs such as *C. perfringens* ε-toxin. *S. aureus* α-toxin and related toxins contain three domains with a more globular structure than the elongated aerolysin family PFTs. *B. anthracis* protective antigen (PA) is the binding component of *B. anthracis* edema and lethal toxins. PA structure with four domains is similar to that of PFO but PA monomer contains only one β-hairpin forming the β-barrel like β-PFTs other than CDCs. Figures were produced with the program MacPyMOL. Green, receptor binding domain; red, pore-forming domain; blue and yellow, oligomerization domains.

**Figure 8 toxins-16-00182-f008:**
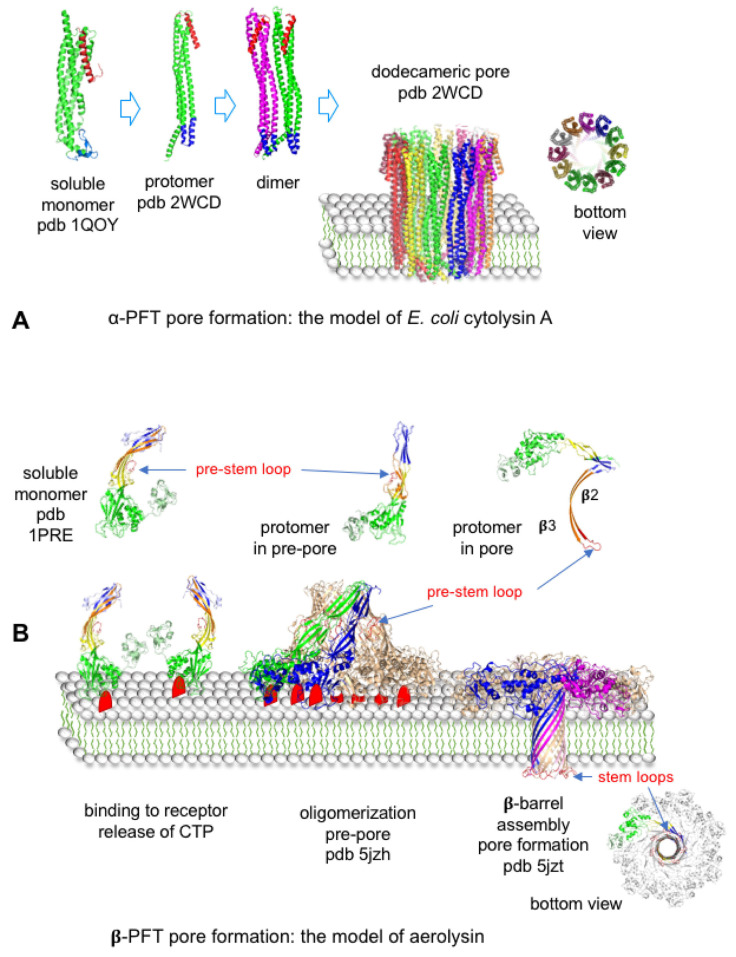
Comparison of pore formation in cell membrane between the α-pore-forming toxin *E. coli* cytolysin A (**A**) and β-pore-forming toxin aerolysin (**B**) [44,160]. Figures were produced with the program MacPyMOL.

#### 4.3.2. Enzymatically Active Toxins against Membrane Compound(s)

Bacteria, notably fermentative bacteria such as clostridia, secrete various hydrolytic enzymes (proteases, phospholipases, carbohydrases, nucleases, hyaluronidases, sialidases, urease, etc.). These enzymes degrade environmental organic substrates into smaller size compounds such as peptides, monosaccharides, and nucleotides, which are transported into bacteria for their metabolism. Certain hydrolytic enzymes are potent toxins involved in tissue degradation, colonization, and dissemination of pathogens in the host [161].

##### Bacterial Phospholipases and Sphingomyelinases

Phospholipases are the most predominant bacterial toxins active on cell membranes through enzymatic activity. A representative toxin of this class of toxins is the *C. perfringens* α-toxin, which is a 43 kDa single-chain protein containing two structural domains. The α-helical N-terminal domain contains the active site and the β-sandwich C-terminal domain is the receptor binding domain [162] (Figure 9). *C. perfringens* α-toxin is the major toxin involved in *C. perfringens* gangrene and acts synergistically with PFO [25,163,164]. The C-terminal domain binds to packed phospholipids of intact membranes in a Ca^++^-dependent manner. This induces conformational changes of loops 1 and 2, which are on each side of the zinc-binding enzymatic site, thus leading to an active conformation of the toxin [162,165]. *C. perfringens* α-toxin is a zinc phospholipase C that preferentially cleaves phosphatidylcholine, yielding phosphorylcholine and diacylglycerol. In addition, sphingomyelin is cleaved into phosphorylcholine and ceramide. *C. perfringens* α-toxin is the first bacterial toxin that has been identified to display an enzymatic activity [166]. At high concentrations, *C. perfringens* α-toxin degrades membrane phosphatidylcholine and sphingomyelin leading to membrane disruption and cell lysis. Thereby, *C. perfringens* α-toxin induces myonecrosis and hemolysis. Moreover, diacylglycerol and ceramide, which are released by sublytic doses of *C. perfringens* α-toxin, activate various signal transduction pathways including activation of endogenous phospholipases A2, C, and D as well as protein kinase C, thus leading to multiple cellular effects. Upregulation of the intercellular adhesion molecule 1 (ICAM-1) and endothelial leukocyte adhesion molecule 1 (ELAM-1) causes the accumulation of neutrophils in blood vessels and subsequent thrombosis. Impaired neutrophil migration to infected tissues facilitates the growth and dissemination of *C. perfringens*. Indeed, *C. perfringens* gangrene is characterized by the absence of neutrophil infiltration in infected tissues. In addition, activation of the arachidonic cascade generates thromboxane A2, leukotrienes, and prostaglandins, which promote local inflammation and vasoconstriction. Increased production of interleukin-8, TNFα, and platelet-activating factor (PAF) are responsible for local inflammation with increased vascular permeability, edema, and platelet aggregation [167,168,169,170]. Passage of *C. perfringens* α-toxin and PFO in the blood circulation causes intravascular hemolysis, cardiovascular collapse, shock, and organ failure [171,172,173].

Various bacterial pathogens produce either bacterial wall-associated or secreted phospholipases/sphingomyelinases, which contribute to their pathogenicity (Table 2) [174,175,176,177].

##### Collagenases and Proteases

Certain pathogens secrete potent and specific collagenases and proteases, which contribute to their pathogenesis. Thereby, *Clostridium histolyticum* is responsible for severe myonecrosis and gangrene and secrete collagenases as major toxins. *C. histolyticum* produces at least seven forms of collagenases (68 to 130 kDa), which process from two protein precursors encoded by two distinct genes. *C. histolyticum* collagenases are generated by proteolytic removal of C-terminal fragments and belong to two classes, namely class I (ColG) and class II (ColH). Despite the low level of amino acid sequence identity, *C. histolyticum* collagenases share a common structural organization with a C-terminal receptor-binding domain and an N-terminal enzymatic domain containing the zinc-dependent protease motif HExxH. *C. histolyticum* attacks all collagen types and gelatin through multiple cleavages at PxGP motifs in the collagen triple helix, thus causing severe destruction of connective tissue [178,179,180,181,182,183,184].

*C. perfringens* secretes a collagenase, ColA (or κ-toxin) related to ColG, hyaluronidase, and sialidase, which likely contribute to the gangrene lesions [183,185].

Certain *Bacteroides fragilis* strains secrete an enterotoxin and are responsible for acute and chronic intestinal diseases in children and adults as well as in young animals. In addition, enterotoxigenic *B. fragilis* plays a role in colorectal cancer [186,187]. *B. fragilis* enterotoxin (BFT) is a 21 kDa zinc-dependent protease that exists in three closely related isoforms. BFT is secreted as an inactive protoxin, which is activated by proteolytic removal of an N-terminal 170 amino acid domain. The catalytic C-terminal domain (190 amino acids) contains the zinc protease motif HExxH and shows structural similarity with the adamalysin/ADAM protease family from eukaryotes [188]. BFT cleaves the extracellular domain of E-cadherin leading to impaired intercellular junctions, morphological changes of intestinal epithelial cells, and increased mucosal permeability [189,190]. Cleavage of the E-cadherin extracellular domain induces the release of β-catenin, which links the E-cadherin intracellular domain to actin filaments. Nuclear translocation of β-catenin upregulates the c-myc pathway leading to cell proliferation. BFT-induced E-cadherin cleavage stimulates signaling pathways such as MAPKs and subsequently NF-κB, resulting in increased secretion of inflammatory cytokines (IL-8), increased fluid secretion, and inflammatory responses. However, BFT promotes host defenses in the late period after exposure including activation of autophagy and delayed apoptosis, as well as negative regulation of the NF-κB pathway by accumulation of β-catenin in nuclei in the late period. Thus, long-term carriage of enterotoxigenic *B. fragilis* exposes to the risk of chronic intestinal disease and oncogenic transformation mediated by activation of pro-carcinogenic and inflammatory cascades including c-myc, NF-κB, and IL-17 [187,188,189,191,192,193,194].

## 5. Bacterial Protein Toxins Active Intracellularly

Bacterial protein toxins, which are active intracellularly, contain at least three functional domains: a receptor binding domain, translocation domain, and catalytic domain, which is released into the cytosol. Single-chain protein toxins such as DT and clostridial neurotoxins are cleaved between the catalytic domain (A) and the rest of the molecule by bacterial or eukaryotic proteases in the external medium. Both toxin fragments remain linked by a disulfide bridge, which is reduced by the thioredoxin–thioredoxin reductase system for the passage through the endosomal membrane [195,196]. Single-chain protein toxins from the large clostridial glucosylating toxins (LCGTs) such as *C. difficile* TcdA and TcdB contain a cysteine protease domain (CPD), which cleaves the catalytic domain from the rest of the molecule during the translocation of A through the endosomal membrane. The C-terminal part of LCGTs contains repetitive sequences (combined repetitive oligopeptides, CROPs), which in closed conformation prevent CPD-dependent autoprocessing, whereas the open conformation is suitable for toxin binding to the receptor and CPD interaction [197,198,199]. In the AB_5_ toxins, the A fragment is proteolytically cleaved into A1 and A2, which are linked by a disulfide bridge. AB_5_ toxins use a long endocytic pathway and the A1 fragment translocates into the cytosol through the ER-associated protein degradation (ERAD) pathway. Protein disulfide isomerase (PDI) dissociates the A1 domain from A2, which links A1 to the B pentamer, and unfolds A1 for its transport to the cytosol [200,201]. Binary toxins translocate the enzymatic component into the cytosol through channels formed by the oligomerized binding components in the endosomal membrane (see above). Thereby, bacterial protein toxins use different strategies to protect and transport the intracellularly active domain into the cytosol: toxin activation by proteolytic cleavage in the external medium and a disulfide bridge between catalytic and transport domains, which are reduced in the cellular translocation compartment, or autoproteolytic cleavage of the catalytic domain under control by a dynamic C-terminal domain. Thus, translocation of the A domain either through pores in the endosomal membrane (binary toxins), T domain-mediated chaperone mechanism (single-chain protein toxins), or using the ER secretion pathway (AB_5_ toxins) requires protein unfolding. This is achieved by acidic pH and/or host cell chaperones such as heat shock proteins Hsp70 and disulfide bridge-reducing proteins. Then, refolding into the active A domain conformation in the cytosol is assisted by host cell chaperones such as Hsp90, peptidyl-prolyl cis-trans isomerase of the cyclophilin, and FK506-binding protein families [196,200,202,203]. Structures of representative bacterial protein toxins that are active intracellularly are shown in Figure 10.

### 5.1. Toxins Inducing Cell Death

Cell killing is a major effect of bacterial toxins. Various strategies are used by bacterial toxins to induce cell death.

Diphtheria toxin (DT) is one of the most potent cytolethal toxins. DT is produced by *Corynebacterium diphtheriae*, the causative agent of a pharyngeal infection called diphtheria. DT is a single-chain protein of 58 kDa. The C-terminal receptor binding domain interacts with the transmembrane glycoprotein, heparin-binding epidermal growth factor precursor (proHB EGF), and drives DT into endocytic vesicles. DT is cleaved between the A and T domains by the cell protease, furin. Both fragments remain linked by a disulfide bridge. Acidification of EEs induces a conformational change in DT, allowing insertion of the central T domain, which is constituted of 10 α-helices, into the endosomal membrane and subsequent translocation of the N-terminal catalytic (A) domain into the cytosol. The A domain catalyzes the ADP-ribosylation of the elongation factor-2 (EF-2). Thus, DT cleaves NAD and transfers the ADP(adenine diphosphate)-ribose to a diphthamide (post-translationally modified histidine) at position 699 of EF-2 (Figure 11 and Figure 12), leading to inactive EF-2 and subsequent inhibition of protein synthesis and cell death. It is assumed that a single DT molecule can kill one cell [204,205,206,207].

*Pseudomonas aeruginosa* exotoxin A (ExoA) shares a similar mechanism of action with DT despite a different structure organization. ExoA is a single-chain protein of 68 kDa with three functional domains: A, T, and B. In contrast to DT, the receptor-binding domain (B) is in the N-terminal region and the A domain is in the C-terminal position. ExoA receptor is the α2-macroglobulin receptor/low-density lipoprotein-receptor-related protein (LRP). ExoA enters cells via receptor-mediated endocytosis, is activated by furin, and is transported from EEs to the Golgi and then ER. The A domain is delivered into the cytosol through Sec63. Similarly to DT, ExoA ADP-ribosylates diphthamide 699 of EF-2 and blocks the protein synthesis [208,209,210].

Shiga toxin (ST) is produced by *Shigella dyssenteriae* and ST-like toxins (STLs) by enterohemorrhagic *E. coli* such as *E. coli* O157. ST and STLs retain an AB_5_ structure. The ST pentamer of B subunits recognizes globotriaosylceramide Gb3 as a receptor on target cells. ST is endocytosed and transits through the EE, Golgi, and ER. In the *trans*-Golgi network, the A domain is proteolytically cleaved into A1 (27.5 kDa), which contains the enzymatic site and the linker A2 (4.5 kDa); both domains remain linked by a disulfide bridge. In the ER, the disulfide bond is reduced and A1 is translocated into the cytosol through the ERAD pathway. The ST domain A1 is an RNA *N*-glucosidase that cleaves 28S ribosomal RNA, leading to inhibition of protein synthesis and cell death [211,212] (Figure 12).

Anthrax lethal toxin is a binary toxin constituted of PA and LF (see above). The LF structure shows four domains, the N-terminal domain 1 binds to PA oligomers, domains 2 and 3 are involved in substrate recognition, and domain 4 contains the zinc-dependent metalloprotease site. PA mediates the internalization of LF into the cytosol (see above). LF cleaves MAPKKs resulting in decreased gene transcription and subsequent cell death. Notably, LF induces apoptosis of macrophages and dendritic cells [85,213,214,215,216,217].

**Figure 11 toxins-16-00182-f011:**
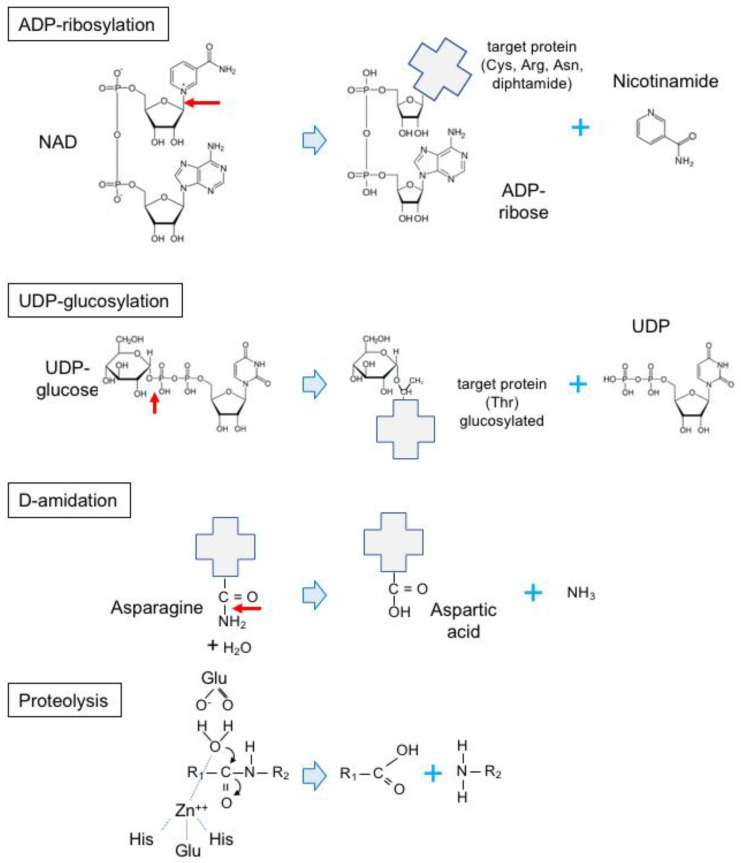
Representative enzymatic activities mediated by intracellularly active bacterial toxins: ADP-ribosylation, glucosylation, de-amidation, and zinc-dependent proteolysis. NAD, nicotinamide adenine dinucleotide; ADP, adenine diphosphate; UDP, uridine diphosphate glucose; Cys, cysteine; Arg, arginine; Asn, asparagine; Thr, threonine; Glu, glutamic acid; His, histidine.

Cytolethal distending toxins (CDTs), also called genotoxins, are tripartite toxins including two binding components (CdtA, 25 kDa, and CdtC, 20 kDa) and one enzymatic domain (CdtB, 30 kDa). The CDT nomenclature does not correspond to “A” enzymatic domain and “B” receptor-binding domain. CDTs are produced by more than 30 bacterial pathogens, mostly Gram-negative bacteria, such as *E. coli*, *Campylobacter jejuni*, *Haemophilus ducreyi*, *S. dyssenteriae*, and *Aggregatibacter actinomycetemcomitans*. The three proteins assemble in the bacterial periplasm and are secreted as a mature holotoxin. CdtA and CdtC interact with glycan structures harbored by a broad range of gangliosides and *N*-linked glycoproteins and mediate the toxin endocytosis. CdtB undergoes a retrograde transport through the Golgi and ER. CdtB probably uses the ERAD pathway to exit from the ER and enter the nucleus thanks to its nuclear localization sequences. CdtB shares a similar structure with DNase I and cuts DNA but with a lower efficiency than DNase I. This results in cell cycle arrest and cell death by apoptosis, notably in lymphocytes and monocytes, which are highly susceptible to CDT intoxication. Certain cells can survive by activating DNA repair mechanisms. However, this can induce genomic instability with enhanced mutation frequency and capacity to grow in an anchorage-independent manner leading to carcinogenesis [218,219,220,221] (Figure 12).

*Helicobacter pylori* is a Gram-negative bacterium responsible for chronic gastritis which can evolve into severe diseases such as peptic ulcer, lymphoma, and gastric adenocarcinoma. One of the main virulence factors is the vacuolating toxin VacA. VacA is synthesized as a 140 kDa prototoxin that undergoes proteolytic N- and C-terminal cleavages and is secreted as a mature 88 kDa protein. VacA is further cleaved into N-terminal 33 kDa and C-terminal 55 kDa fragments, which remain non-covalently linked. VacA oligomerizes in solution and disassembles at acidic pH. It is proposed that VacA monomers bind to cell membrane unknown receptor(s), oligomerize, insert into the membrane, and accumulate intracellularly. VacA develops multiple cellular effects such as vacuolation, autophagy, activation of p38 MAPK, increased cell membrane permeability, and paracellular permeability. A prominent effect consists of cell death by apoptosis through alteration of the mitochondria membrane. VacA targets mitochondria and forms channels in the mitochondrial membrane stimulating the release of cytochrome *c* and other mitochondrial proteins into the cytosol leading to apoptosis. In addition, VacA activates the proapoptotic factors BAX and BAK [222,223,224] (Figure 12).

**Figure 12 toxins-16-00182-f012:**
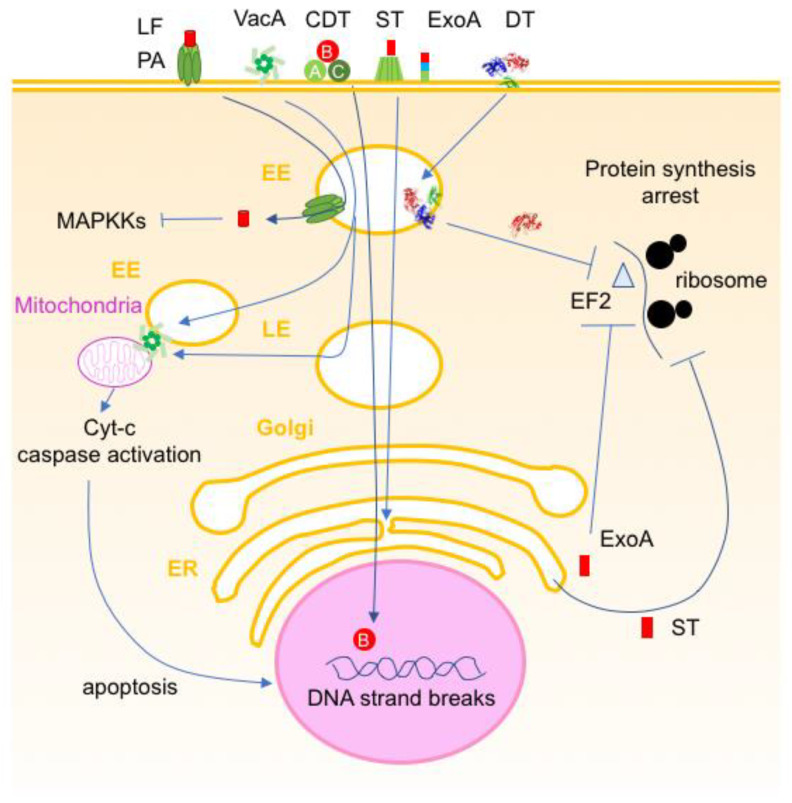
Schematic representation of intracellularly active bacterial toxins inducing cell death. Diphtheria toxin (DT) and *Pseudomonas aeruginosa* exotoxin A (ExoA) deliver their enzymatic domains (A) via early endosomes (EE) and endoplasmic reticulum (ER) pathways, respectively, into the cytosol and inactivate the elongation factor 2 (EF2) by ADP-ribosylation. Shiga toxin (ST) uses a long entry pathway through ER and hydrolysis ribosomal RNA bonds. *Bacillus anthracis* lethal toxin delivers the lethal factor (LF) via protective antigen (PA) pores through the endosomal membrane and proteolytically cleaves mitogen-activated protein kinase kinases (MAPKKs). Cytolethal distending toxins (CDT) translocate the enzymatic components CdtB into the nucleus, which has DNase activity. *Helicobacter pylori* vacuolating toxin (VacA) oligomerizes and is internalized via EEs, which can contact mitochondria and transfer the toxin to this organelle. VacA is also delivered from late endosomes (LE) [224]. VacA permeabilizes the mitochondrial membrane leading to the release of cytochrome *c* (Cyt-c) and apoptosis.

### 5.2. Toxins Perturbing Cell Homeostasis

Certain bacterial toxins modulate host cell signaling pathways without inducing cell death. This is the case notably for various bacterial enterotoxins which induce active secretion of water and electrolytes from enterocytes.

Cholera toxin (CT) is synthesized by *Vibrio cholerae*, the causative agent of cholera. CT and heat-labile (LT) enterotoxins from enterotoxigenic *E. coli*, which induce cholera-like diarrheas, share a similar AB_5_ structure with that of ST. The B pentamer of CT and LTs binds to GM1 gangliosides that are localized in lipid rafts of cell membranes such as in enterocytes. The interaction of CTB subunits with multiple GM1 molecules with particular ceramide structures induces membrane curvature and further toxin endocytosis via clathrin-dependent or clathrin-independent pathways according to the cell type. CTA1 is the catalytic domain and is linked to the CTB pentamer through the CTA2 polypeptide by non-covalent bonds. CTA is activated by proteolytic cleavage between CTA1 and CTA2, with both chains remaining linked by a disulfide bridge. The C-terminal sequence, KDEL of CTA2, enhances the retrograde transport from EE, through the trans-Golgi network, and then to the ER. The disulfide bond between CTA1 and CTA2 is reduced and the ER chaperone protein disulfide isomerase facilitates the CTA1 dissociation from the rest of the toxin. Dissociated CTA1 spontaneously unfolds and uses the ERAD pathway with both Sec61 and Hrd1 translocons to enter the cytosol. CTA1 refolds with the help of the cytosolic chaperone Hsp90 and binds to the ADP-ribosylation factor ARF6, which activates CTA1 by inducing an active conformational change. Then, CTA1 ADP-ribosylates (Figure 11 and Figure 13) the α subunit of the stimulatory heterotrimeric G-protein (Gsα) at Arg201 leading to inhibition of the intrinsic GTPase activity of Gsα, which remains bound to GTP and thus in permanently in an active conformation. Continuous stimulation of adenylate cyclase by CT ADP-ribosylated Gsα results in a massive increase in intracellular cAMP levels and subsequent activation of protein kinase A, which phosphorylates the cystic fibrosis transmembrane conductance regulator (CFTR). This leads to the active secretion of Cl^−^ and water (Figure 13) [225,226,227,228].

Pertussis toxin (PTX) is one of the main toxins produced by *B. pertussis*, which is responsible for whooping cough. PTX belongs to the AB_5_ toxin family with one catalytic subunit (S1, 28 kDa) and five B subunits. However, in contrast to CT and ST, the PTX B pentamer consists of four distinct proteins: S2 (23 kDa), S3 (22 kDa), S4 (11.7 kDa, two copies), and S5 (9.3 kDa). PTX recognizes glycoconjugates as receptors and thus targets a wide variety of cells. PTX undergoes receptor-mediated endocytosis and follows a retrograde transport from EE to ER through the Golgi similarly to CT and ST. In the ER, S1 dissociates from the B pentamer and translocates into the cytosol through the ERAD pathway. PTX is an ADP-ribosyltransferase that targets the inhibitory heterotrimeric G-protein Giα at Cys-351 in the C-terminus. ADP-ribosylation (Figure 13) of Giα in its C-terminal domain impairs the interaction with its effectors, thus yielding inactive Giα. Adenylate cyclase, which no longer receives negative signals, massively catalyzes the formation of cAMP leading to stimulation of cAMP-dependent signaling pathways through PKA. In addition, PTX mediates ADP-ribosylation-independent effects through the interaction of the B pentamer with receptors such as lymphocytosis, inhibition of chemotaxis, modulation of innate and adaptive immune response including stimulation of dendritic cells and T cells, and production of pro-inflammatory cytokines (Figure 13) [229,230].

Anthrax edema toxin uses the same mechanism for cell entry as anthrax lethal toxin. PA internalizes the edema factor (EF) through pores in the membrane of acidified endosomes. EF (89 kDa) is an adenylate cyclase, which, as the intrinsic eukaryotic adenylate cyclase, promotes the elevation of cAMP levels. This results in the stimulation of regulatory pathways controlled by cAMP through PKA and exchange protein activated by cAMP (Epac). A prominent effect in the host is the production of edema and vascular dysfunctions (Figure 13) [214,217].

**Figure 13 toxins-16-00182-f013:**
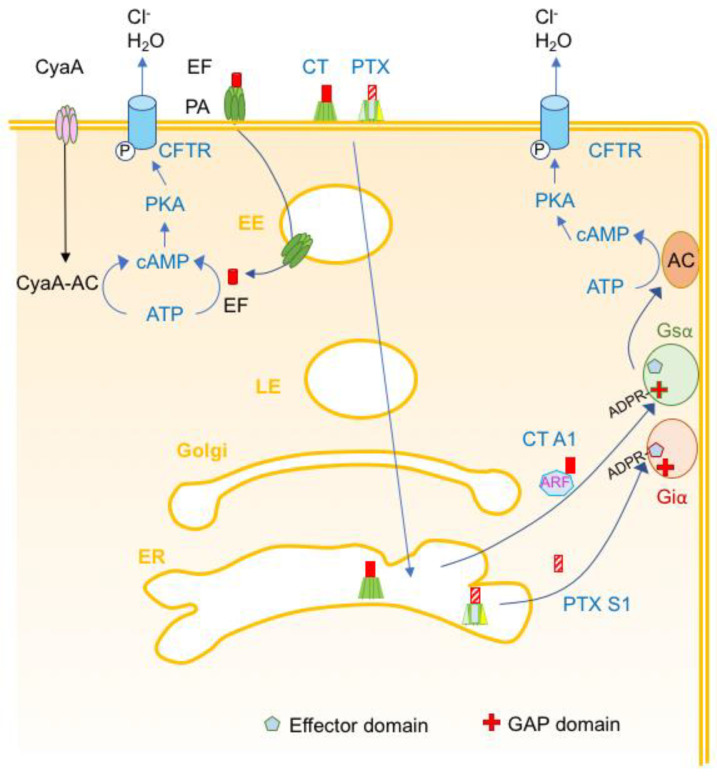
Schematic representation of intracellularly active bacterial toxins that modify cell homeostasis. Cholera toxin (CT) and Pertussis toxin (PTX) deliver their enzymatic domains, CT-A1 and PTX-S1, respectively, via early endosome (EE), late endosome (LE), Golgi, and endoplasmic reticulum (ER). CT-A1 binds to the ADP-ribosylating factor (ARF) and ADP ribosylates the stimulatory heterotrimeric Gs α-subunit at Arg-201 in the GAP (GTPase activating protein) domain yielding permanent active Gsα that stimulates the eukaryotic adenyl-cyclase (AC). This results in excessive formation of cAMP and subsequent activation of the cystic fibrosis transmembrane conductance regulator (CFTR) and excessive efflux of Cl^−^ and H_2_O. PTX S1 ADP ribosylates the inhibitory heterotrimeric Gi α-subunit at Cys-351 in the effector interaction domain leading to inactivation of Giα. Thus, AC only receives stimulatory signals from Gsα resulting in excessive formation of cAMP. Anthrax edema toxin delivers the edema factor (EF) through pores mediated by the oligomerized protective antigen (PA) in the endosomal membrane. *B. pertussis* adenylate cyclase (CyaA) forms pores in the plasma membrane and delivers the CyaA-AC domain into the cytosol. The bacterial AC domains, EF and CyaA-AC, catalyze the formation of cAMP from ATP that subsequently activates CFTR.

### 5.3. Toxins Targeting the Actin Cytoskeleton

Actin is one of the most abundant proteins in all cells. Actin is in a dynamic equilibrium between monomers and filaments, which polymerize at one end and disassemble at the opposite extremity. The actin cytoskeleton contributes to cell morphology and various cell functions such as locomotion, endocytosis, exocytosis, trafficking of intercellular organelles, and maintenance of intercellular junctions. Various bacterial toxins target the actin cytoskeleton directly or indirectly.

#### 5.3.1. Actin ADP-Ribosylating Toxins

Clostridial binary toxins (*C. perfringens* Iota toxin, *C. botulinum* C2 toxin, *C. difficile* binary toxin or transferase, and *C. spiroforme* toxin) and related binary toxins of *Bacillus cereus* and *B. thuringiensis* (vegetative insecticidal proteins) share a similar organization with *B. anthracis* toxins consisting of binding and enzymatic components. Binding components recognize a cell surface receptor (lipolysis-stimulated lipoprotein receptor, LSR, for Iota toxin and *C. difficile* transferase, and asparagine-linked carbohydrates for C2 toxin), form heptamers, and trap the corresponding enzymatic components. Binary toxins are receptor-mediated endocytosed. Acidification of EE triggers the insertion of oligomers into the EE membrane and the formation of functional pores allowing the transfer of the enzymatic components into the cytosol. Translocation of the enzymatic component also requires cytosolic chaperones [158,231,232,233,234]. The intracellular activity of clostridial and related *Bacillus* binary toxins consists of monoADP-ribosylation of actin monomers at Arg177 that is located in the interface between actin monomers. This results in the inhibition of actin polymerization at the plus or growing end of actin filaments while the depolymerization at the minus end is preserved. Whereas *C. perfringens* Iota toxin, *C. difficile* transferase, and *C. spiroforme* toxin ADP-ribosylates all actin isoforms, C2 toxin only modifies β/γ non-muscle and γ-smooth muscle actin isoforms. Thereby, intoxicated cells show a complete disorganization of the actin cytoskeleton, with actin being totally in its monomeric form. The impairment of the cytoskeleton alters the integrity of endothelial and epithelial barriers leading to an increased permeability. These toxins induce necrotic and hemorrhagic enteritis in animals. *C. difficile* transferase is an additional virulence factor in *C. difficile* infections in humans [158].

*Photorhabdus luminescens* is a Gram-negative entomopathogenic bacterium that colonizes the gut of nematodes and produces several toxins targeting the actin cytoskeleton. PTC3 is a toxin complex containing the binding component TcdA1 that retains a pentameric structure, the linker TcdB2 component, and the enzymatic TccC3 component. TccC3 is an ADP-ribosyltransferase that modifies all isoforms of actin at Thr148, as well as in the monomeric form and assembled in filaments. ADP-ribosylation at Thr148 impairs the binding to thymosin-*β*4, which is a sequestering actin protein, and prevents actin polymerization. Therefore, ADP-ribosylation at Thr148 induces an opposite effect to that of Arg177 by promoting actin polymerization instead of inhibition of actin filament assembly [235] (Figure 14).

#### 5.3.2. Toxins Modifying Rho-Family GTPases

C3 exoenzyme is the pioneer molecule that was found to target Rho-GTPases and allowed to unravel their important role in the regulation of the actin cytoskeleton [236,237,238]. C3 is produced by *C. botulinum* types C and D strains in addition to the neurotoxins C1 and D, respectively, and C2 toxin. C3 is also synthesized by *Clostridium limosum*, *Bacillus cereus*, and *S. aureus* (C3stau or EDIN, epithelial differentiation inhibitor) strains. C3 exoenzymes are 25–28 kDa proteins that share a similar structure with the enzymatic components of clostridial binary toxins. In contrast to the clostridial binary toxins that target monomeric actin, C3 exoenzymes specifically modify Rho GTPases. No binding component has been associated to C3 exoenzymes. These proteins have been designated exoenzymes instead of toxins since they cannot efficiently enter cells. However, C3 proteins are efficiently internalized into macrophages, monocytes, and dendritic cells via EE and acidification by a non-yet-determined receptor(s) and endocytic mechanism [239,240,241]. *C. botulinum* C3 ADP-ribosylates RhoA, B, and C at Asn41 are located in the switch I region that undergoes conformational changes between the active and inactive state. When activated by guanine nucleotide exchange factors (GEFs), Rho translocates to the membrane where it interacts with the downstream effectors stimulating actin filament polymerization. ADP-ribosylated Rho remains in the cytosol in complex with guanine nucleotide dissociation inhibitor (GDI). Thus, the signaling of actin polymerization is blocked by C3 ADP-ribosylation of Rho. C3 induces cell rounding, loss of actin stress fibers, and inhibition of various cell functions linked to the actin cytoskeleton such as locomotion, endocytosis/phagocytosis, exocytosis, and cell-cycle progression [242,243] (Figure 14).

*C. difficile* toxin A (TcdA) and toxin B (TcdB) are part of the large clostridial glucosylating toxin (LCGT) family that also encompasses *Paeniclostridium sordellii* (previously *Clostridium sordellii*) toxins (hemorrhagic toxin, TcsH, and lethal toxin, TcsL), *Clostridium novyi* α-toxin (TcnA), and toxin *C. perfringens* large cytotoxin (TpeL). *C. difficile* is responsible for antibiotic-associated diarrhea, enteritis, and pseudomembranous colitis, whereas *P. sordellii* and *C. novyi* are mainly associated with gas gangrene in humans and animals. In addition, *P. sordellii* causes toxic shock syndrome, notably in women, and enterotoxemia in cattle [244,245,246]. LCGTs are 250–300 kDa single-chain proteins that contain four functional domains: a C-terminal receptor binding domain characterized by combined repetitive oligopeptide sequences (CROPS), an N-terminal glucosyltransferase domain, an autoprotease domain, and a central translocation domain (Figure 3). Receptors have been identified for TcdB as frizzled receptors in epithelial colonocytes and chondroitin sulfate proteoglycan 4 on other cell types such as myofibroblasts, sulfated glucosaminoglycans and low-density lipoprotein receptor (LDLR) for TcdA and TcnA, semaphorins (promoting angiogenesis in vascular endothelial cells) for TcsL, and LDL receptor-related protein 1 (LRP1) for TpeL [247,248,249,250,251,252,253]. LCGTs enter cells via receptor-mediated endocytosis and the catalytic domain, which is processed by autoproteolytic cleavage, is released into the cytosol from acidified EEs [254]. LCGTs inactivate small GTP-binding proteins by monoglucosylation at Thr35/37 using UDP-glucose as a sugar donor (Figure 10) [8,255,256,257,258]. Glucosylation at Thr35/37, which is located in the switch I, impairs the interaction with the downstream effectors of the small GTP-binding proteins. TcdB and TcdA glucosylate Rho family proteins (RhoA, B, C, Rac, and Cdc42), whereas TcsL modifies mainly Rac from the Rho family and in addition Ras proteins. TcnA catalyzes the transfer of glucosamine from UDP-*N*-acetylglucosamine. The major consequences are an alteration in the actin cytoskeleton and disorganization of the intercellular junctions [8,259,260]. Moreover, TcdB and TcdA induce a strong inflammatory response through inactivation of RhoA and activation of pyrin inflammasome [261], whereas TcsL triggers apoptosis of immune and phagocytic cells through inactivation of Ras. TcsL causes drastic disorganization of focal adhesions, actin cytoskeleton, and intercellular junctions leading to massive edema [262,263,264,265].

Another class of bacterial toxins modifies Rho family proteins but in their active conformation. Cytotoxic necrotizing factors (CNF1, 2, and 3) from *E. coli*, mainly uropathogenic strains, CNFγ from *Yersinia pseudotuberculosis*, and dermonecrotic toxin from *Bordetella bronchiseptica* are part of the deamidating toxin group. CNF1 is the prototype of this toxin family. CNF1 is a 115 kDa single-chain protein with an N-terminal receptor binding domain and a C-terminal catalytic domain. CNF1 interacts with the laminin receptor and the Lutheran adhesion glycoprotein/basal cell adhesion molecule (Lu/BCAM) and enters the cell via receptor-mediated endocytosis [266,267]. The acidification of endosomes promotes the translocation of the proteolytically cleaved C-terminal domain into the cytosol [268]. CNF1 catalyzes the deamidation of Gln61/63 of Rho, Rac, and Cdc42 into glutamic acid (Figure 10). The conserved Gln61/63 is located in the GTPase domain and deamidation of Gln61/63 results in an irreversible GTP-bound conformation of Rho proteins and thus permanent active molecules. A major consequence of Rho GTPases is a drastic reorganization of the actin cytoskeleton with increased formation of stress fibers and lamellipodia and increased phagocytosis and cell migration but destabilization of epithelial barrier integrity. CNF1 induces additional multiple cellular effects such as the induction of proinflammatory cytokines IL-1 and IL-6 and the inflammatory response, cell cycle arrest in the G2/M phase, blockade of apoptosis, or induction of apoptosis according to the cell type and toxin concentration. However, the effects of CNF are transient since activated Rho GTPases are ubiquitinylated and degraded by the proteasome [268,269,270] (Figure 14).

*P. luminescens* TccC5 also activates Rho GTPases by targeting Gln61/63 but through ADP-ribosylation instead of deamidation [271].

A comparison of the mode of action of bacterial toxins that alter the actin cytoskeleton by targeting either the actin or Rho-GTPase molecules is shown in Figure 14.

### 5.4. Neurotoxins Impairing Neurotransmitter Release

A particular class of bacterial toxins are the clostridial neurotoxins, which specifically target neuronal cells and impair the release of neurotransmitters, namely botulinum neurotoxins (BoNTs) and tetanus neurotoxin (TeNT). BoNTs are responsible for botulism, which is characterized by flaccid paralysis, whereas TeNT induces spastic paralysis, the tetanus. BoNTs are divided into eight toxinotypes (A to H) and are produced by diverse *Clostridium* species (*C. botulinum* groups I, II, III, *Clostridium argentinense*, neurotoxigenic strains of *Clostridium butyricum*, and *Clostridium baratii*), while TeNT is a more homogenous group of toxins produced by *Clostridium tetani* [272,273]. Clostridial neurotoxins are synthesized as single-chain proteins of 150 kDa that are activated by a proteolytic cleavage between the light chain (LC) N-terminal 50 kDa domain containing the enzymatic site and the 100 kDa heavy chain (HC) that consists of two C-terminal receptor binding domains and a N-terminal translocation domain. LC and HC remain linked by a disulfide bridge. As receptors, BoNTs and TeNT recognize both a ganglioside, notably GT1b and GD1a, and a specific membrane protein of neuronal cells, synaptic vesicle protein 2 (SV2) for BoNT/A, E, D, F, and TeNT, which also uses the basal lamina protein nidogen and synaptotagmin for BoNT/B, D/C, and G. In contrast to TeNT, BoNTs associate with non-toxic proteins (non-toxic non-hemagglutinin (NTNH) protein, hemagglutinins, or OrfX proteins) to form botulinum complexes that are resistant to acidic pH and protease degradation. Thus, BoNT complexes are stable through their passage in the digestive tract. BoNTs and TeNT enter motor neuron endings via receptor-mediated endocytosis. BoNTs act in the peripheral nervous system, whereas TeNT undergoes a retrograde transport to the central nervous system and targets inhibitory interneurons. LC is released from acidified endosomes and through disulfide bond reduction. BoNTs and TeNT are zinc-dependent metalloproteases that cleave SNARE (soluble *N*-ethylmaleimide-sensitive factor attachment protein receptor) complex proteins including SNAP25, synaptobrevin, and syntaxin, which have a crucial role in docking and fusion of synaptic vesicles containing neurotransmitters with the presynaptic membrane. Thus, BoNTs block the release of acetylcholine from motor neuron endings and TeNT inhibits that of glycine and GABA (gamma-aminobutyric acid) from central inhibitory interneurons [274,275,276,277].

## 6. Insights into the Evolution of Bacterial Protein Toxins

Bacterial protein toxins are unique molecules secreted by certain bacteria that are able to recognize specific target eukaryotic cells and to act at the cell membrane or to deliver an intracellularly active domain. They are among the most powerful molecules and are active at very low concentrations. Thereby, bacterial toxins are responsible for severe diseases such as botulism, diphtheria, and tetanus. Striking characteristics of bacterial protein toxins are their wide diversity regarding their size, structure, and mode of action. The edition of bacterial toxins is a dynamic process. Most bacterial toxins are part of toxin families that contain multiple isoforms or variants according to the bacterial strains. Bacteria exploit various genetic mechanisms such as gene duplication, mutation, insertion/deletion, and recombination to generate novel or modified proteins. Toxin genes are located on diverse genetic support chromosomes, phages, plasmids, and transposons, which facilitate horizontal gene transfer between bacteria [161,278]. For example, 695 protein sequences have been identified for CDT subunits from at least 32 Gram-negative bacterial species. Based on phylogenetic analysis, CDT sequences show a high level of diversity, the catalytic subunit CdtB being the most conserved. It has been speculated that *Yersinia* CDTs represent the most ancestral lineage with subsequent horizontal transfer to other bacterial species [279]. It is noteworthy that the main habitat of toxigenic bacteria is the environmental soil, dust, and sediments, including the intestinal tract of healthy humans and animals where these bacteria can transit. The high density and diversity in bacteria in the intestinal tract (microbiota) facilitate genetic exchanges between bacteria. Horizontal gene transfer might also concern exchanges between bacteria and host eukaryotic cells. Indeed, sequences of *cdtB* and gene-encoding STB subunits have been found in the genome of gall midge. Phylogenetic analysis supports gene transfer from bacteria to insects. It is suggested that toxin genes might confer a protective function in developing larvae and pupae against other insect predators such as wasps [280].

Intriguing questions concern the origin of bacterial toxins and their evolution. Single-chain protein toxins represent the simplest structures and might represent the first bacterial toxins. Some of these toxins could originate from hydrolytic enzymes, which are secreted by certain bacteria such as fermentative bacteria. While hydrolytic enzymes have a broad spectrum of activity, toxins have developed an enzymatic activity for specific substrate(s). For example, BoNTs, *B. anthracis* LF, and *B. fragilis* BFT retain the enzymatic site HExxH, which is common in many metalloproteases. However, they target specific substrates, SNARE proteins for BoNTs, MAPKKs for LF, and E-cadherin for BFT. Another possibility is that certain bacterial toxins could have been horizontally transferred from mammalian cells to bacteria as it has been suggested for BFT based on structure similarity [188].

For optimal activity, bacterial toxins have acquired a modulable structure by differentiation, recombination, insertion, and/or deletion of specific domains (catalytic, translocation/delivery, autoprocessing, and receptor-binding domains) involved in the different steps of intoxication. For example, modular recombination has been documented for CNFs, *Bordetella* dermonecrotic toxins, and related toxins [278]. In *C. difficile*, the extensive diversification of TcdB seems to be mainly based on recombination events, with independent evolution of the four functional domains [281,282]. Kumar and Singh suggest that molecular flexibility and dynamics of protein toxins drive the evolutionary processes. Protein plasticity is an important aspect of toxins, including the transition from a disordered structure that facilitates toxin transport through cell membranes to an ordered structure required for specific interaction with receptor/substrate. Co-evolution of bacterial toxins and substrates as well as induced folding from disordered to ordered structures that enhances adaptation and affinity to substrate or receptor could be a crucial step in bacterial toxin evolution [283].

More intriguing is the emergence of complex structures resulting from the combination of distinct proteins to form a functional toxin. Multiprotein toxins are essentially intracellularly active toxins. Separation of the receptor-binding/delivery module from the catalytic subunit provides the advantage of long-lasting internalization process of the toxin, since the receptor-binding/delivery module can be recycled to the cell membrane to trap additional catalytic subunits. For example, binding components of *C. perfringens* iota-toxin are re-exposed to the cell surface, even to the opposite cell side of administration, and internalize the enzymatic component [284]. A similar transcytotic pathway has been found for cholera toxin [285]. The assembly of multiprotein toxins is achieved through the insertion of domain or linker sequences in the enzymatic components that interact with the oligomerized receptor-binding components. A typical example is provided by C3 enzyme and clostridial binary toxins such as C2 and Iota toxins. C3 is a 25 kDa protein with core β-strands containing the ADP-ribosylation site surrounded by α-helices (Figure 14). The C2 enzymatic component (C2-I) is a 50 kDa protein that is divided into two structural subdomains of 25 kDa. Both C2-I subdomains share a similar folding that is related to that of the C3 enzyme despite a low amino acid sequence identity. The C-terminal C2-I subdomain retains the ADP-ribosylation activity towards actin monomers instead of Rho GTPase for C3 and the N-terminal subdomain is required for interaction with C2 oligomerized binding components and internalization into the cytosol. It is conceivable that the C2 gene is derived from an ancestral C3 gene by duplication and adaptation to substrate or interacting molecule(s) [161]. In contrast, AB_5_ toxins (CT, ST, and PTX) use a short peptide (A2 peptide) located in the C-terminal part of the A fragment as a linker between the A1 subunit and B pentamer (see above).

## 7. Concluding Remarks

Recurrent questions are still how and for which purpose bacteria have developed such diverse and such potent toxin molecules. Indeed, bacterial toxins are powerful pathogenicity factors. A relevant example is provided by toxigenic bacteria involved in gangrene, such as clostridia. The production of potent toxins that induce massive tissue destruction and a blockade of host defenses allows a rapid progression of bacteria in host tissues. The habitat of these bacteria is usually the environment where they can grow and survive for long periods. They do not need to attack and kill a host to survive. Rather, they have a role in the decomposition of cadavers. Thus, bacterial gangrenes likely emerge as accidental events of the interaction of environmental bacteria and a host. Regarding the high diversity in bacterial enterotoxins, is it the result of the coevolution of bacteria and host intestines that benefits bacteria? It has been suggested that, by modulating bacterial and host metabolism, CT facilitates fecal–oral transmission of *V. cholerae* [286]. However, the emergence and evolution of toxigenic bacteria not only occurs in human or animal host environments but also in soil, aquatic, or plant environments under various selective pressures and genetic exchanges between bacterial populations and phages [278]. More enigmatic are the bacterial neurotoxins that recognize specific neuronal cells and target the specialized machinery of neurotransmission. The origin and benefit for bacteria to produce such neurotoxins are unclear. Indeed, non-toxigenic counterparts of *C. botulinum* or *C. tetani* survive and spread in the environment as well as toxigenic strains indicating that toxin production is not essential for the regular life of these bacteria [287,288]. In addition to being harmful factors, bacterial toxins are the basis of various useful applications. Their modular structure enables the engineering of recombinant efficient and specific tools, notably for therapeutic purposes. The receptor binding domain can be adapted to interact with a specific cell population and can be combined with a catalytic domain able to kill the target cells or to modulate a cellular process. Currently, the most widely used therapeutic toxin is the native BoNT/A, which is used for an increasing number of applications from neurological disorders, ophthalmology, gastro-enterology, urology, dermatology, pain, depression, and cosmetics [275,289]. Another relevant toxin application concerns insecticidal toxins produced by *Bacillus thuringiensis* and related *Bacillus* sp. including parasporal crystal protein toxins (Cry), cytotoxins (Cyt), and vegetative insecticidal proteins (VIP). *B. thuringiensis* is widely used as spores to control insect proliferation in agriculture [159,290].

## Figures and Tables

**Figure 1 toxins-16-00182-f001:**
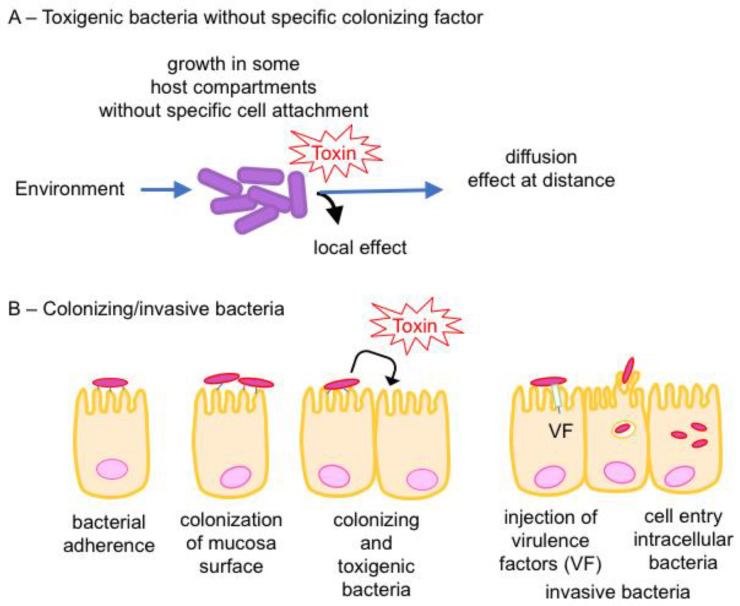
Toxigenic (**A**) and colonizing/invasive (**B**) bacteria. Toxigenic bacteria secrete toxins that act locally and at a distance from the site of infection, while colonizing/invasive bacteria interact directly with the target cells, some of them produce potent toxins or inject virulence factors (VF) into cells.

**Figure 2 toxins-16-00182-f002:**
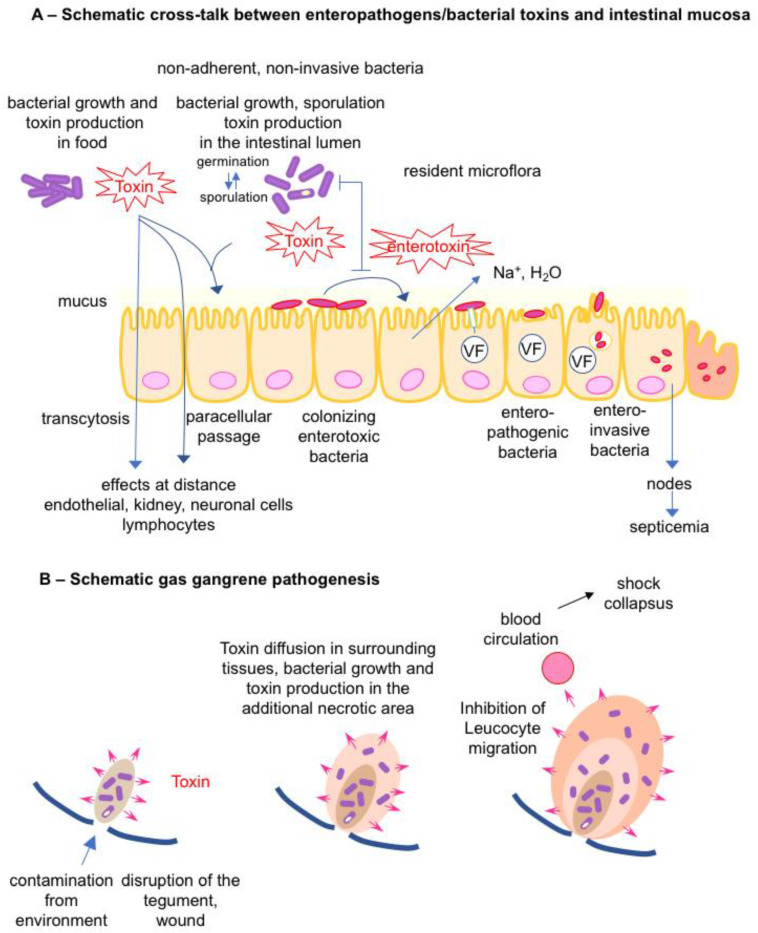
Intestinal (**A**) and gas gangrene (**B**) schematic models of infection with toxigenic bacteria. (**A**) Toxigenic bacteria grow and produce toxins either in food, intestinal lumen, or on the intestinal mucosa surface. Entero-invasive bacteria inject virulence factors (VF) into intestinal cells, which mediate their cell entry. (**B**) Toxigenic bacteria enter into the organism through tegument breaking. The presence of necrotic tissues facilitates initial bacterial growth and production of toxin(s), which diffuse locally leading to additional tissue necrosis and subsequent bacterial growth and toxin synthesis. In addition, toxins impair host defenses via inhibition of leucocyte/macrophage migration and inhibition of phagocytosis. Passage of sufficient amounts of toxins into the blood circulation leads to toxic shock.

**Figure 5 toxins-16-00182-f005:**
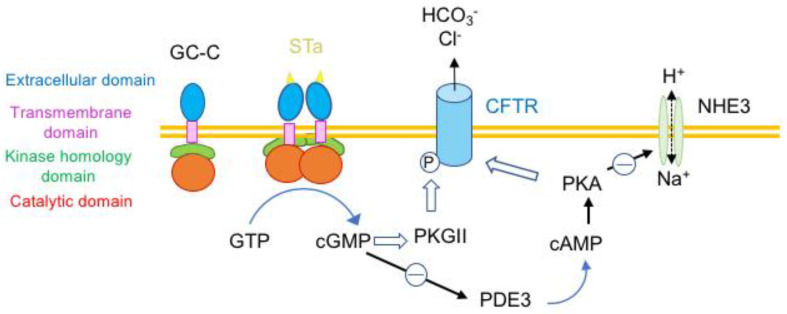
Schematic representation of the *Escherichia coli* thermostable enterotoxin (STa), a hormone-like toxin. STa binds to guanylyl cyclase C (GC-C) and induces GC-C dimerization and activation through conformational change in the inhibitor kinase homology domain. Increased level of cGMP activates cGMP-dependent protein kinase II (PKGII) and subsequently the cystic fibrosis transmembrane conductance regulator (CFTR) channel. In addition, cGMP inhibits the phosphodiesterase 3 (PDE3) resulting in the accumulation of cAMP and subsequently, further activation of CFTR and inhibition of the Na^+^/H^+^-exchanger 3 channel (NHE3).

**Figure 6 toxins-16-00182-f006:**
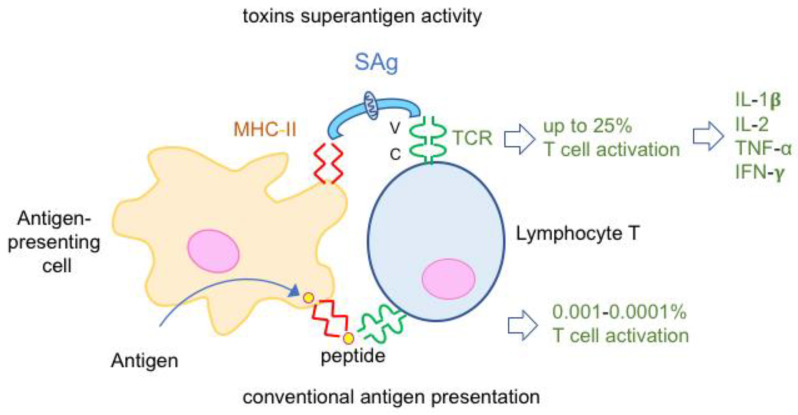
Superantigen toxins (SAg) bind to both MHC-II molecules on the antigen-presenting cell and TCR variable region (v) on lymphocyte T leading to up to 25% T cell activation and release of pro-inflammatory cytokines (IL-1β, IL-2, IL-6, and TNF-α) and interferon-γ (IFN-γ). In contrast, the conventional antigen presentation activates only a low number of T lymphocytes. v, variable; c, constant regions.

**Figure 9 toxins-16-00182-f009:**
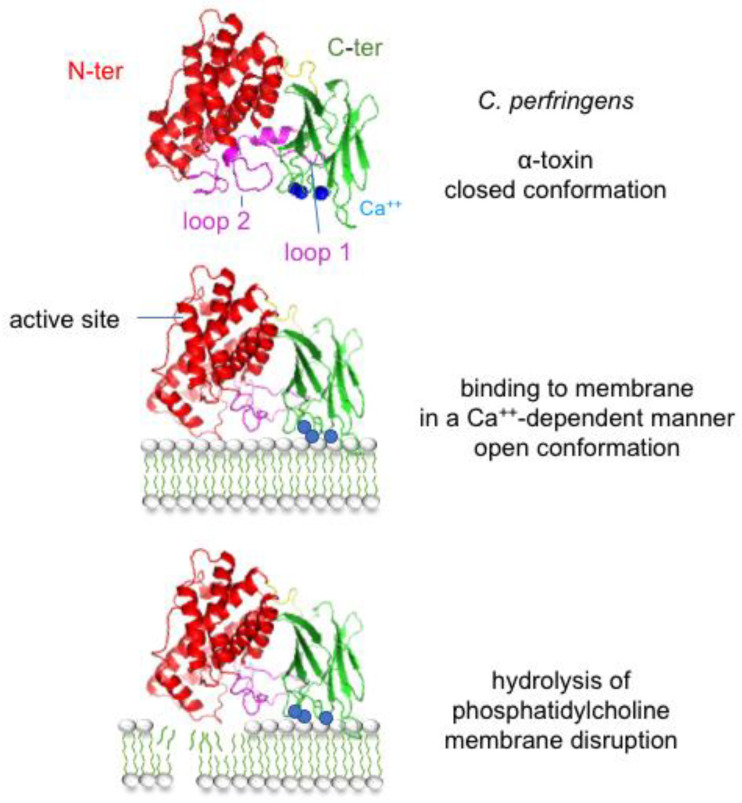
Interaction of *C. perfringens* α-toxin with the cell membrane. *C. perfringens* α-toxin is secreted as an inactive conformation. Binding to membrane phospholipid in a Ca^++^-dependent manner induces a conformational change in loops 1 and 2, resulting in active toxin conformation. Hydrolysis of phosphatidylcholine and sphingomyelin at high toxin concentrations leads to membrane disruption. Inactive conformation, pdb 1QMD, and active conformation, pdb 1CA1; blue spheres, Ca^++^. Figures were produced with the program MacPyMOL.

**Figure 10 toxins-16-00182-f010:**
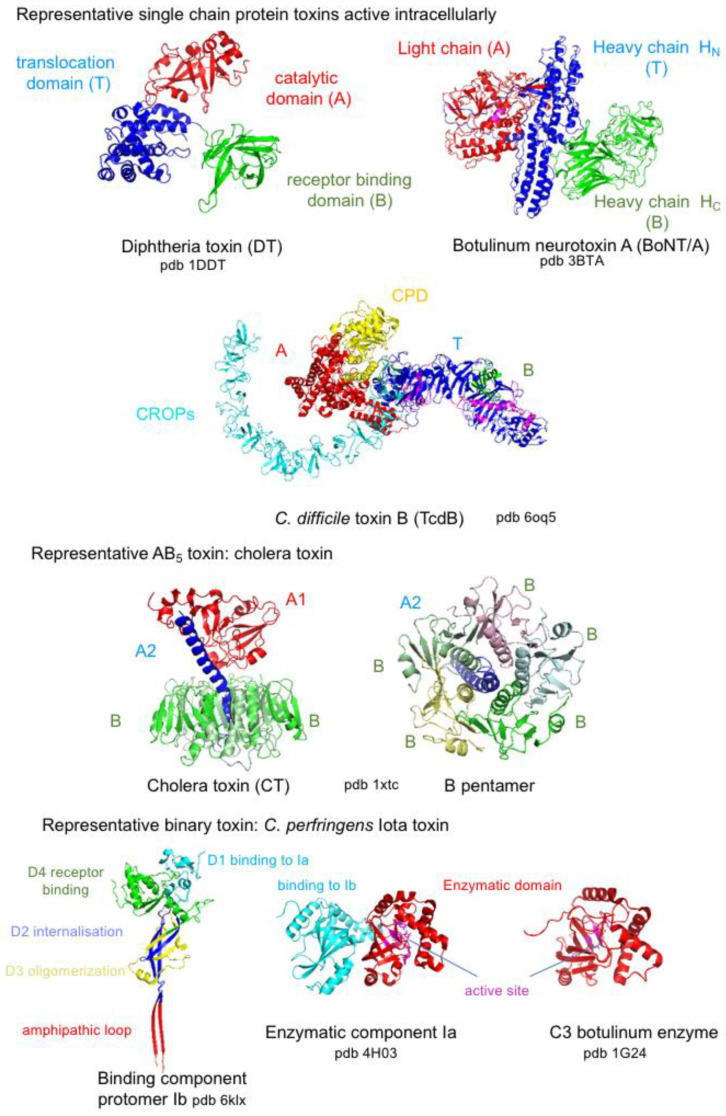
Structures of representative intracellularly active toxins from single-chain protein AB_5_ and binary toxin families. The TcdB structure corresponds to the open conformation. The enzymatic component of Iota toxin contains two structurally related domains: an interactive domain with a binding component and an enzymatic domain that shares a similar structure with that of the C3 enzyme. C3 lacks a receptor binding structure and binding component. CPD, cysteine protease domain; CROPs, combined repetitive oligopeptides. Figures were produced with the program MacPyMOL.

**Figure 14 toxins-16-00182-f014:**
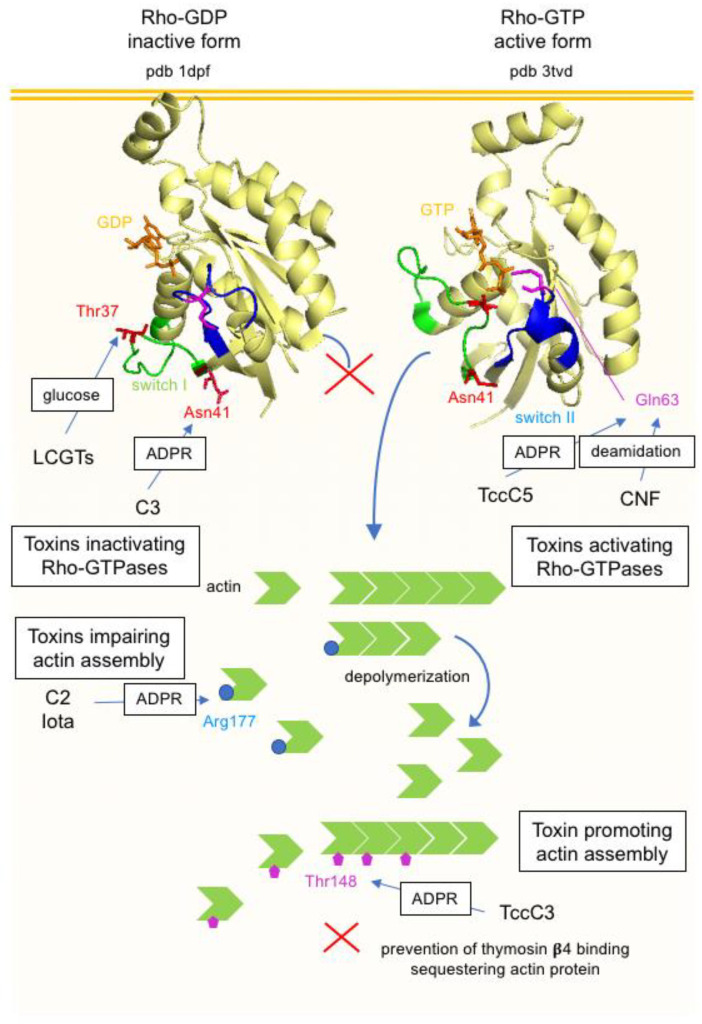
Schematic representation of the intracellularly active bacterial toxins that alter the actin cytoskeleton. Bacterial toxins either attack actin molecules or Rho-GTPases. Clostridial binary toxins such as C2 and Iota toxins ADP-ribosylate actin monomers at Arg177 in the actin-actin binding site thus prevent actin filament assembly, while *Photorhabdus luminescens* TccC3 ADP-ribosylates actin at Thr148 preventing the association with the negative regulator thymosin-b4 and thus promotes polymerization of actin. Bacterial toxins such as the C3 enzyme and large clostridial glucosylating toxins (LCGTs) modify a residue in the switch I of Rho-GTPases that is essential for the interaction with their downstream effectors and thus block the Rho-GTPases in the inactive form linked to GDP. In contrast, other bacterial toxins (deamidating toxins and TccC5) alter Gln61/63 in the switch II, which is involved in the hydrolysis of GTP in GDP, leading to permanently active Rho-GTPases and increased actin polymerization.

**Table 2 toxins-16-00182-t002:** Representative bacterial phospholipases and sphingomyelinases that contribute to virulence.

Bacteria	Toxin/Enzyme	Enzyme Class	Substrate Specificity	Role in Pathogenicity
*Bacillus anthracis*	SMase PI-PLC	SMC, PLC	SM, PI, PC	macrophage associated growth of *B. anthracis*, bacterial escape from phagosome
*Bacillus cereus*	SMase	SMC	SM	evasion of macrophage
	PLC	PLC	PC, PE, PS	hemolysis
*Clostridium novyi*	γ-toxin	PLC	PC, PE, PI, LPC, PG, SM	myonecrosis
*Clostridium perfringens*	α-toxin	PLC, SMC	PC, PE, PI, PG, PS, SM	gas gangrene
*Corynebacterium pseudotuberculosis*		SMD	SM, LPC	platelet aggregation, intravascular coagulation, endothelial hyperpermeability, bacterial spreading
*Helicobacter pylori*		SMC	SM	apoptosis in gastric cells
	PldA1	PLA2	PC	hemolysis, bacterial survival and growth at low pH
*Legionella pneumophila*	PlaB	PLA2	PC, PG	hemolysis, destruction of lung surfactant
	VipD/PatA	PLA2	PC, LPC	establishment of bacterial niche
*Listeria monocytogenes*	PLC-B	PLC, SMC	PC, PE, PS, SM	bacterial escape from phagosome bacterial cell to cell spreading
	PLC-A	PLC	PI	
*Mycobacterium tuberculosis*	PLC-A, PLC-B, PLC-C, PLC-D	acid phosphatases	PC, SM	pulmonary infection, necrosis of alveolar macrophages
*Pseudomonas aeruginosa*	PLC-H	acid phosphatases	PC, LPC, PE, PG, SM	cytotoxic to human monocytes and endothelial cells, hemolysis, platelet aggregation and thrombosis
*Pseudomonas aeruginosa*	ExoU	PLA2	PC, PE, PA, PI	focal adhesion disruption, cytoskeleton alteration, inflammatory response
*Salmonella enterica* serovar Typhimurium	SseJ	PLA1 and GCATase		endosomal tabulation and establishment of bacterial intracellular niche
*Staphylococcus aureus*	β-toxin	SMC	SM, LPC	bacterial survival in neutrophils, cytotoxic to neutrophils and monocytes
	PI-LPI	PLC	PI, LPI	bacterial survival in neutrophils
*Streptococcus pyogenes*	SlaA	PLA2	PC, PE, PS, LPC	bacterial adherence to host cells, cytotoxicity

PC, phosphatidylcholine; PE, phosphatidylethanolamine; PG, phosphatidylglycerol; PI, phosphatidylinositol; PS, phosphatidylserine; PLC, phospholipase C; LPC, lysophosphatidylcholine; LPI, lysophosphatidylinositol; SM, sphingomyelin; SMC sphingomyelinase C; SMD, sphingomyelinase D; GCATase, cholesterol acyl transferase.

## Data Availability

Data is contained within the article.

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
