# Peer review of "Overview of Bacterial Protein Toxins from Pathogenic Bacteria: Mode of Action and Insights into Evolution"

_toxins, 2024, doi:10.3390/toxins16040182_

Round 1

Reviewer 1 Report

Comments and Suggestions for Authors

Comprehensive data pertaining to the bacterial protein toxins generated by toxigenic bacteria was incorporated in this review. Insights into the evolution of bacterial protein toxins and numerous examples of such toxins accompanied by their diverse modes of action are presented in the review. Readers in the field would benefit from the critical insights that the review has furnished regarding bacterial protein toxins.

Below are some comments for this review:

1. The title should emphasize the bacterial protein toxins from toxigenic bacteria since there are also many other bacterial protein toxins that are produced from non-pathogenic bacteria and do not cause any diseases in humans or animals.

2. Figure 4 lacks clarity in its illustration. The precise mechanism by which receptors participate in endocytosis or influence cell membrane activity remains obscure. The descriptions of the short and lengthy pathways are ambiguous. This requires further clarification.

3. Figure 8 lacks a description of how the dodecameric ClyA pore is assembled and forms pores.

Some minor points: some words need correction, such as can grow (not can growth).

Author Response

Comprehensive data pertaining to the bacterial protein toxins generated by toxigenic bacteria was incorporated in this review. Insights into the evolution of bacterial protein toxins and numerous examples of such toxins accompanied by their diverse modes of action are presented in the review. Readers in the field would benefit from the critical insights that the review has furnished regarding bacterial protein toxins.

Below are some comments for this review:

  1. The title should emphasize the bacterial protein toxins from toxigenic bacteria since there are also many other bacterial protein toxins that are produced from non-pathogenic bacteria and do not cause any diseases in humans or animals.

The title has been modified accordingly "Overview of Bacterial Protein Toxins from Pathogenic Bacteria". The term pathogenic was used to avoid repetition with bacterial toxins.

  1. Figure 4 lacks clarity in its illustration. The precise mechanism by which receptors participate in endocytosis or influence cell membrane activity remains obscure. The descriptions of the short and lengthy pathways are ambiguous. This requires further clarification.

Figure 4 has been modified to further clarify the short and long endocytic pathways. The contribution of toxin interaction with receptors on mechanisms of endocytosis has been further documented. At the end of paragraph 2, the following text has been added:

Interaction between bacterial toxin and cell receptor not only drives the cell specificity of toxins but also promotes membrane curvature that is required for membrane invagination and formation of endocytic vesicles. Most of toxins which enter cell via a short pathway recognize cell membrane protein as receptor and use a clathrin-dependent endocytosis. Polymerization of clathrin associated to activity of accessory proteins triggers membrane invagination and formation of clathrin-coated vesicles. Bacterial toxins which use a long pathway such as cholera toxin (CT) and Shiga toxin (ST) interacts with glycosphingolipids as receptors, GM1 and globotriaosylceramide, respectively (see below). Their pentameric structure of B components bind to multiple receptor molecules leading to membrane lipid reorganization and clustering of toxin-receptor complexes which promote tubular membrane invagination [31].

The sentence " Interaction of intracellularly active toxins with cell surface receptor not only drives the cell specificity but also initiates membrane curvature and endocytic vesicle formation" has been added in the legend of Fig. 4.

  1. Figure 8 lacks a description of how the dodecameric ClyA pore is assembled and forms pores.

 Detailed description of dodecameric ClyA pore formation has been included in Fig. 8.

Some minor points: some words need correction, such as can grow (not can growth).

These mistakes have been corrected. The text has been checked again for grammatical errors.

Reviewer 2 Report

Comments and Suggestions for Authors

This paper is a review of overall bacterial toxins. It is a broad theme, and it is difficult to summarize all of the aspects of bacterial toxins. However, this review covers the broad theme well. I have some comments and suggestions that help understand general toxins readers.

Table 1

It summarizes oligomer numbers and pore size. It is better to mention the structural papers of these pores (not prepared) as a reference. Some references include these structural papers. So please update it and include these papers as references.

PA:

Atomic structure of anthrax protective antigen pore elucidates toxin translocation.

Jiang J, Pentelute BL, Collier RJ, Zhou ZH.Nature. 2015 May 28;521(7553):545-9. doi: 10.1038/nature14247. Epub 2015 Mar 16.

Iota: Cryo-EM structures reveal translocational unfolding in the clostridial binary iota toxin complex.

Yamada T, Yoshida T, Kawamoto A, Mitsuoka K, Iwasaki K, Tsuge H. Nat Struct Mol Biol. 2020 Mar;27(3):288-296. doi: 10.1038/s41594-020-0388-6. Epub 2020 Mar 2. PMID: 32123390

CDTb: Structural insights into the transition of Clostridioides difficile binary toxin from prepore to pore.

Anderson DM, Sheedlo MJ, Jensen JL, Lacy DB. Nat Microbiol. 2020 Jan;5(1):102-107. doi: 10.1038/s41564-019-0601-8. Epub 2019 Nov 11.

Figure 7

The author would like to show prepore, pores, or monomers?

It is better to summarize pore structures.

It should be one of the representative structures of each group in Table 1.

In Table 1, the author showed six groups, which is why there should be six representative structures.

Figure 8

Does the author want to show the representative b-PFT pore formations from monomer and prepore.

If so, it may be better to exchange Figure 7 and Figure 8.

Author Response

This paper is a review of overall bacterial toxins. It is a broad theme, and it is difficult to summarize all of the aspects of bacterial toxins. However, this review covers the broad theme well. I have some comments and suggestions that help understand general toxins readers.

Table 1

It summarizes oligomer numbers and pore size. It is better to mention the structural papers of these pores (not prepared) as a reference. Some references include these structural papers. So please update it and include these papers as references.

PA:

Atomic structure of anthrax protective antigen pore elucidates toxin translocation.

Jiang J, Pentelute BL, Collier RJ, Zhou ZH.Nature. 2015 May 28;521(7553):545-9. doi: 10.1038/nature14247. Epub 2015 Mar 16.

Iota: Cryo-EM structures reveal translocational unfolding in the clostridial binary iota toxin complex.

Yamada T, Yoshida T, Kawamoto A, Mitsuoka K, Iwasaki K, Tsuge H. Nat Struct Mol Biol. 2020 Mar;27(3):288-296. doi: 10.1038/s41594-020-0388-6. Epub 2020 Mar 2. PMID: 32123390

CDTb: Structural insights into the transition of Clostridioides difficile binary toxin from prepore to pore.

Anderson DM, Sheedlo MJ, Jensen JL, Lacy DB. Nat Microbiol. 2020 Jan;5(1):102-107. doi: 10.1038/s41564-019-0601-8. Epub 2019 Nov 11.

 The references in Table 1 have been updated according to your suggestions

Figure 7

The author would like to show prepore, pores, or monomers?

It is better to summarize pore structures.

It should be one of the representative structures of each group in Table 1.

In Table 1, the author showed six groups, which is why there should be six representative structures.

Figure 7 has been revised showing structures of monomers, protomers and pores of representative toxins of each group indicated in Table 1. 

Figure 8

Does the author want to show the representative b-PFT pore formations from monomer and prepore.

If so, it may be better to exchange Figure 7 and Figure 8.

Figure 8 compares pore formation between a- and b-PFT with CyaA as model of a-PFT and aerolysin as model of b-PFT. Pore formation by CyaA has been further detailed. It seems clearer to have Figure 7 (structures of monomers, protomers, pores) and then Figure 8 (pore formation of representative a- and b-PFT). If requested, Figures 7 and 8 can be exchanged.

Round 2

Reviewer 2 Report

Comments and Suggestions for Authors

This paper is a review of overall bacterial toxins. It is a broad theme, and it is difficult to summarize all aspects of bacterial toxins. However, this review covers the broad theme well. I have some comments and suggestions that help understand general toxins readers.
Table 1
It summarizes oligomer numbers and pore size. It is better to mention the structural papers of these pores (not prepared) as a reference. Some references include these structural papers. So please update it and include these papers as references.
PA:
Atomic structure of anthrax protective antigen pore elucidates toxin translocation.
Jiang J, Pentelute BL, Collier RJ, Zhou ZH.Nature. 2015 May 28;521(7553):545-9. doi: 10.1038/nature14247. Epub 2015 Mar 16.
Iota: Cryo-EM structures reveal translocational unfolding in the clostridial binary iota toxin complex.
Yamada T, Yoshida T, Kawamoto A, Mitsuoka K, Iwasaki K, Tsuge H. Nat Struct Mol Biol. 2020 Mar;27(3):288-296. doi: 10.1038/s41594-020-0388-6. Epub 2020 Mar 2. PMID: 32123390
CDTb: Structural insights into the transition of Clostridioides difficile binary toxin from prepore to pore.
Anderson DM, Sheedlo MJ, Jensen JL, Lacy DB. Nat Microbiol. 2020 Jan;5(1):102-107. doi: 10.1038/s41564-019-0601-8. Epub 2019 Nov 11.
The references in Table 1 have been updated according to your suggestions
In Table 1 (ver 2), I see no changes in the references.
Please check it again.

Binary toxin
ref 85(lipid membrane), 151(LSR) for Ib
ref 151(LSR) for CDTb
ref 152 and 153 for VIP2
In VIP2, is this really experimetally detremined 7mer? Pkease check  the paper.

Figure 7
The author would like to show prepore, pores, or monomers?
It is better to summarize pore structures.
It should be one of the representative structures of each group in Table 1.
In Table 1, the author showed six groups, which is why there should be six representative structures. Figure 7 has been revised showing structures of monomers, protomers and pores of representative toxins of each group indicated in Table 1.
> I am still confused. According to Table 1, the author can show all representative structures as follows. a-PFA(cytolysin A, RTX, and 3d Cry toxins) and b-PFT(CDCs, Aerolysin, a-toxins, and binary toxin). The author classified monomer (fig 7A) and monomer in complex (fig 7B). These are different. If the author wants to show the pro-form monomer, please show these monomers. Moreover, which part is pro-region. If the author wants to show the monomer in the complex, the author should mention it from pre-pore or pore. It is still confusing, so it should be summarized in this figure of this review.

Figure 8
Does the author want to show the representative b-PFT pore formations from monomer and prepore?
If so, it may be better to exchange Figure 7 and Figure 8.
Figure 8 compares pore formation between - and -PFT with CyaA as model of a-PFT and aerolysin as model of b-PFT. Pore formation by CyaA has been further detailed. It seems clearer to have Figure 7 (structures of monomers, protomers, pores) and then Figure 8 (pore formation of representative - and -PFT). If requested, Figures 7 and 8 can be exchanged.
>The author showed CyaA and aerolysin as the representative a-PFT and b-PFT. The order is acceptable.
Pore formation by CyaA has been further detailed. > This figure was revised? Or was the detailed pore formation mechanism unknown in a PFT?

Author Response

I am very sorry. The original version has been loaded instead of the revised version. Please find the correct revised version.

It summarizes oligomer numbers and pore size. It is better to mention the structural papers of these pores (not prepared) as a reference. Some references include these structural papers. So please update it and include these papers as references.
PA:
Atomic structure of anthrax protective antigen pore elucidates toxin translocation.
Jiang J, Pentelute BL, Collier RJ, Zhou ZH.Nature. 2015 May 28;521(7553):545-9. doi: 10.1038/nature14247. Epub 2015 Mar 16.
Iota: Cryo-EM structures reveal translocational unfolding in the clostridial binary iota toxin complex.
Yamada T, Yoshida T, Kawamoto A, Mitsuoka K, Iwasaki K, Tsuge H. Nat Struct Mol Biol. 2020 Mar;27(3):288-296. doi: 10.1038/s41594-020-0388-6. Epub 2020 Mar 2. PMID: 32123390
CDTb: Structural insights into the transition of Clostridioides difficile binary toxin from prepore to pore.
Anderson DM, Sheedlo MJ, Jensen JL, Lacy DB. Nat Microbiol. 2020 Jan;5(1):102-107. doi: 10.1038/s41564-019-0601-8. Epub 2019 Nov 11.
The references in Table 1 have been updated according to your suggestions
In Table 1 (ver 2), I see no changes in the references.
Please check it again.

R – It was a mistake in the reference formating. Now the suggested references have been correctly included.

Binary toxin
ref 85(lipid membrane), 151(LSR) for Ib
ref 151(LSR) for CDTb
ref 152 and 153 for VIP2
In VIP2, is this really experimetally detremined 7mer? Pkease check  the paper.

The number of protomers in VIP pore has not been exactly determined. It is supposed to be 7 according to structure similarity with heptameric PFT. Thus the number of monomers is indicated as 7?

The references of binary toxins in Table 1 have been checked

Figure 7

> I am still confused. According to Table 1, the author can show all representative structures as follows. a-PFA(cytolysin A, RTX, and 3d Cry toxins) and b-PFT(CDCs, Aerolysin, a-toxins, and binary toxin). The author classified monomer (fig 7A) and monomer in complex (fig 7B). These are different. If the author wants to show the pro-form monomer, please show these monomers. Moreover, which part is pro-region. If the author wants to show the monomer in the complex, the author should mention it from pre-pore or pore. It is still confusing, so it should be summarized in this figure of this review.

Figure 7 has been modified. Fig 7A shows the structure of soluble monomers of representative alpha-PFTs. Detailed structures of the different states are not available for all these toxins.

Figure 7B shows the structure of soluble monomers, protomers from pore complex, and pores of representative toxins from each beta-PFT family.

Pore formation by CyaA has been further detailed. > This figure was revised? Or was the detailed pore formation mechanism unknown in a PFT?

Figure 8 has been modified to show pore formation of representative alpha- and beta-PFT.

Round 3

Reviewer 2 Report

Comments and Suggestions for Authors

I think the revised manuscript is very good for the pore-forming toxin researchers.

But please check the references again.

***

Author's Notes

I am very sorry. The original version has been loaded instead of the revised version. Please find the correct revised version.

It summarizes oligomer numbers and pore size. It is better to mention the structural papers of these pores (not prepared) as a reference. Some references include these structural papers. So please update it and include these papers as references.
PA:
Atomic structure of anthrax protective antigen pore elucidates toxin translocation.
Jiang J, Pentelute BL, Collier RJ, Zhou ZH.Nature. 2015 May 28;521(7553):545-9. doi: 10.1038/nature14247. Epub 2015 Mar 16.
Iota: Cryo-EM structures reveal translocational unfolding in the clostridial binary iota toxin complex.
Yamada T, Yoshida T, Kawamoto A, Mitsuoka K, Iwasaki K, Tsuge H. Nat Struct Mol Biol. 2020 Mar;27(3):288-296. doi: 10.1038/s41594-020-0388-6. Epub 2020 Mar 2. PMID: 32123390
CDTb: Structural insights into the transition of Clostridioides difficile binary toxin from prepore to pore.
Anderson DM, Sheedlo MJ, Jensen JL, Lacy DB. Nat Microbiol. 2020 Jan;5(1):102-107. doi: 10.1038/s41564-019-0601-8. Epub 2019 Nov 11.
The references in Table 1 have been updated according to your suggestions
In Table 1 (ver 2), I see no changes in the references.
Please check it again.

R – It was a mistake in the reference formating. Now the suggested references have been correctly included.

> Based on the version 3 manuscript, there was no changes of the reference of table I.  Three papers, which I suggested, are not included in version 3.

(References were not updated in new table. 1?)

 Figure 7 and Figure 8 are revised. It is very good. But, there are several newly added PDB structures (2BK1, 5jzt, 6rb9, 7AHL, EF6 (mistake?)). Finally, please check again to add these manuscripts to the references.

Binary toxin
ref 85(lipid membrane), 151(LSR) for Ib
ref 151(LSR) for CDTb
ref 152 and 153 for VIP2
In VIP2, is this really experimetally detremined 7mer? Pkease check  the paper.

The number of protomers in VIP pore has not been exactly determined. It is supposed to be 7 according to structure similarity with heptameric PFT. Thus the number of monomers is indicated as 7?

The references of binary toxins in Table 1 have been checked

 >Please check again!!

Author Response

Answer to Reviewer 2

> Based on the version 3 manuscript, there was no changes of the reference of table I.  Three papers, which I suggested, are not included in version 3.

(References were not updated in new table. 1?)

 Figure 7 and Figure 8 are revised. It is very good. But, there are several newly added PDB structures (2BK1, 5jzt, 6rb9, 7AHL, EF6 (mistake?)). Finally, please check again to add these manuscripts to the references.

It was a problem of formatting, now the three additional references are included

Figure 7B, EF6 is wrong, it has been replaced by uze

The other pdb references have been checked and are correct